# Genome-Wide Identification, Structure Characterization, and Expression Pattern Profiling of the Aquaporin Gene Family in *Betula pendula*

**DOI:** 10.3390/ijms22147269

**Published:** 2021-07-06

**Authors:** Jean-Stéphane Venisse, Eele Õunapuu-Pikas, Maxime Dupont, Aurélie Gousset-Dupont, Mouadh Saadaoui, Mohamed Faize, Song Chen, Su Chen, Gilles Petel, Boris Fumanal, Patricia Roeckel-Drevet, Arne Sellin, Philippe Label

**Affiliations:** 1Université Clermont Auvergne, INRAE, PIAF, 63000 Clermont-Ferrand, France; maxime.dupont@etu.uca.fr (M.D.); aurelie.gousset@uca.fr (A.G.-D.); mouadh.saadaoui@etu.uca.fr (M.S.); Gilles.PETEL@uca.fr (G.P.); boris.fumanal@uca.fr (B.F.); patricia.drevet@uca.fr (P.R.-D.); 2Institute of Ecology and Earth Sciences, University of Tartu, 51005 Tartu, Estonia; eele.ounapuu-pikas@ut.ee (E.Õ.-P.); arne.sellin@ut.ee (A.S.); 3National Institute of Agronomy of Tunisia (INAT), Crop Improvement Laboratory, INRAT, Tunis CP 1004, Tunisia; 4Laboratory of Plant Biotechnology, Ecology and Ecosystem Valorization, Faculty of Sciences, University Chouaib Doukkali, El Jadida 24000, Morocco; faizemohamed@yahoo.fr; 5State Key Laboratory of Tree Genetics and Breeding, Northeast Forestry University, 26 Hexing Road, Harbin 150040, China; chensongnet@gmail.com (S.C.); chensu@nefu.edu.cn (S.C.)

**Keywords:** aquaporin, *Betula pendula*, functional diversity, cold stress, RNA-seq, plasticity

## Abstract

Aquaporin water channels (AQPs) constitute a large family of transmembrane proteins present throughout all kingdoms of life. They play key roles in the flux of water and many solutes across the membranes. The AQP diversity, protein features, and biological functions of silver birch are still unknown. A genome analysis of *Betula pendula* identified 33 putative genes encoding full-length AQP sequences (*Bpe*AQPs). They are grouped into five subfamilies, representing ten plasma membrane intrinsic proteins (PIPs), eight tonoplast intrinsic proteins (TIPs), eight NOD26-like intrinsic proteins (NIPs), four X intrinsic proteins (XIPs), and three small basic intrinsic proteins (SIPs). The *Bpe*AQP gene structure is conserved within each subfamily, with exon numbers ranging from one to five. The predictions of the aromatic/arginine selectivity filter (ar/R), Froger’s positions, specificity-determining positions, and 2D and 3D biochemical properties indicate noticeable transport specificities to various non-aqueous substrates between members and/or subfamilies. Nevertheless, overall, the *Bpe*PIPs display mostly hydrophilic ar/R selective filter and lining-pore residues, whereas the *Bpe*TIP, *Bpe*NIP, *Bpe*SIP, and *Bpe*XIP subfamilies mostly contain hydrophobic permeation signatures. Transcriptional expression analyses indicate that 23 *BpeAQP* genes are transcribed, including five organ-related expressions. Surprisingly, no significant transcriptional expression is monitored in leaves in response to cold stress (6 °C), although interesting trends can be distinguished and will be discussed, notably in relation to the plasticity of this pioneer species, *B. pendula*. The current study presents the first detailed genome-wide analysis of the AQP gene family in a *Betulaceae* species, and our results lay a foundation for a better understanding of the specific functions of the *BpeAQP* genes in the responses of the silver birch trees to cold stress.

## 1. Introduction

Aquaporins (AQPs) are small (21 to 34 kDa) channel-forming transmembrane proteins that belong to the major intrinsic protein (MIP) superfamily. They are found in all tissue types, and are localized in the plasma membrane, endomembrane system (endoplasmic reticulum, the Golgi apparatus, lysosomes, vesicles, endosomes, vacuoles), and in the membranes of chloroplasts and mitochondria. They are involved in the bidirectional transfer of water, many small solutes and gases across cell membranes in response to osmotic and hydrostatic pressures, or concentration gradients.

First discovered in animals, AQPs were subsequently found in almost all living organisms [1]. Compared to other living kingdoms, plants harbor a remarkable abundance and divergence of AQP sequences (e.g., 35 AQPs in *Arabidopsis thaliana* [2], 28 in *Beta vulgaris* [3], 40 in *Sorghum bicolor* [4], 52 in *Olea europaea* [5], 54 in *Populus trichocarpa* [6], 71 in *Gossypium hirsutum* [7], and 120 in Canola [8]). This distinctiveness is most probably due to the result of gene duplication and the higher ploidy levels in plants, the higher degree of subcellular compartmentalization of the plant cells, and the need for better water control capacity related to the sessile nature of plants [9]. In this respect, and based on their primary sequence, plant aquaporin homologs are divided into up to five major subfamilies: the plasma membrane intrinsic proteins (PIPs), primarily localized in the plasma membrane; the tonoplast intrinsic proteins (TIPs), predominantly targeted to the vacuolar membrane; the nodulin 26-like intrinsic proteins (NIPs), localized in the plasma membrane and the endoplasmic reticulum; the small basic intrinsic proteins, (SIPs) localized in various endomembrane systems; and the uncharacterized intrinsic proteins (XIPs), mostly related to the plasma membrane [10,11]. Three additional subfamilies also exist: GlpF-like intrinsic proteins (GIPs) and hybrid intrinsic proteins (HIPs) have been described in basal *Plantea* lineage species (*Lycopodiopsida* and *Bryopsida*), and the large intrinsic proteins (LIPs) in the algal phylum Heterokontophyta. These subfamilies are thought to have been lost in vascular plants during the course of evolution, plausibly due to function redundancies [12,13]. Such purging events are not exclusive to these ancestral subfamilies, as evidenced by genome-wide identification and comparison studies on AQPs which have confirmed the loss of the XIPs in monocots, certain plant lineages (*Brassicaceae*) [14,15], and the NIP2s (categorized as NIPIII) in a large group of plant species [5,16].

Despite this high sequence diversity, high resolution coupled to three-dimensional prediction studies of AQPs from different organisms have revealed that the AQP structure displays highly conserved tetrameric characteristics where each protomer resembles an hourglass-like model [17,18,19]. An AQP monomer is typically formed with six trans-membrane (TM) alpha helices (helix 1 to helix 6) connected by five loops (A-E) with both the *N*- and *C*- terminals having a cytoplasmic orientation. This then embeds in the lipid bilayer to make route for water and solute transport inside a so-called pore, described within each monomer.

There are two highly conserved Asn-Pro-Ala (NPA) motifs in two half helices (HB and HE) of loopB and loopE in the middle of the pore. Their steric configuration and distance from each other define the size selectivity barrier for various permeants [20,21]. An additional constriction zone consists of four amino acids known as the “aromatic/Arginine” (ar/R: F58-H182-C191-R197 in AQP1) selectivity filter which defines the substrate specificity and permeability [22]. These residues are located in the TM helix 2 (H2), TM helix 5 (H5), and loop E (LE1 and LE2), respectively.

In addition to these two significant selectivity barriers placed within the pore, several other conserved features are known to play a role in AQP solute specificity, including Froger residues (aka Froger’s positions) (FPs) composed of five conserved amino acids (P1-P5: T116-S196-A200-F212-W213 in AQP1) known to discriminate glycerol-transporting aquaglyceroporins (GLPs) from water-conducting AQPs [23]. Similarly, AQPs exhibit nine specificity-determining positions (SDPs) that were predicted to facilitate the diffusion of some non-aqueous solutes [24]. These motifs, and/or the position of a particular amino acid, are viewed as key elements for the selection and the transport of permeants across the cell membranes.

AQPs play a key role in maintaining the homeostasis of water and solutes by facilitating the transport of water and a variety of inorganic and organic solutes such as *inter alia* boric and silicic acids, ammonia, glycerol, hydrogen peroxide, urea, polyols, glycine, and lactate [25]. The fact that the diffusion of small molecules is of prime importance for the maintenance of cell life implies that the AQPs play major roles in numerous plant physiological processes like reproduction, anther dehiscence, seed germination, fruit ripening, photosynthesis, stomatal regulation, petal- and leaf-water movements, xylem embolism repair, maintenance of cell turgor, and cell elongation [10,26,27]. AQPs are also involved in plant responses to biotic and abiotic environment stress [28]. As a result, huge efforts are being made to assess their role in improving tolerance to abiotic stresses such as drought, salinity, heat, cold, and heavy metal toxicity, as well as in alleviating biotic stresses including the induction of disease immunity pathways with the diffusion of the pathogen-induced apoplastic hydrogen peroxide (H_2_O_2_) to the cytoplasm [29,30,31].

Overall, mining the key AQP genes controlling growth tolerance to various adverse environmental conditions becomes increasingly significant for modern forestry, particularly through high-throughput technologies. To our knowledge, no systemic characterization of the AQP genes has been conducted in *Betula* genus, and the ever-growing availability of plant genomes and transcriptomes (including some *Betula sp.* [32,33]) offers unprecedented access to specific gene families, which can be exhaustively evaluated.

The regulation of the annual cycle of growth and dormancy is of great significance for the growth and survival of boreal and temperate tree species under northern climatic conditions. Air temperature (together with photoperiod) is a main environmental driving-force for this regulation [34], and the cold stress sensing and tolerance deployed by northern tree species, including silver birch, in a context of profound climate change are under investigation [35,36,37]. Cold stress reduces the root (*Lp_r_*) and leaf (*K_l_*_eaf_) hydraulic conductivities [38,39], correlated with a global dehydration [40]. However, cell responses to cold stress and adaptation to the freezing condition are distinct processes overlapped only by some multifaceted interactions of many gene products. It remains that the cold stress response and the water conductivities (i.e., *Lp_r_* and *K*_leaf_) have been connected to the plant’s water uptake mechanism and cell membrane water permeability, i.e., two physiological processes that are tightly regulated by aquaporins.

No study has been conducted on genome-wide identification, sequence characterization (i.e., diversity and structure), or the functional prediction of aquaporins in *Betulaceae*, although some birch genomes have been sequenced and released [32,38,41]. In the present study, the genome of *Betula pendula* was analyzed to identify the aquaporin encoding genes (named *Bpe*AQPs). The genomical, structural, and biochemical features were inferred based on present knowledge for the entire set of *Bpe*AQPs published here from raw and unannotated sequences from NCBI databases, therefore classifying each sequence in the general plant AQP system. Furthermore, the expression profiling of *Bpe*AQPs were analyzed and compared using transcriptome sequencing data available for *B. pendula* from different organs (flower, leaf, xylem, and root), specifically in leaves under cold stress. The knowledge obtained from this study is expected to provide the first comprehensive classification of AQPs in the *B. pendula* species in particular, and in the *Betulaceae* clade in general. In addition, it provides key fundamentals for exploring the functions and mechanisms of *B. pendula* AQP proteins, and their potential roles in physiology and stress alleviation in silver birch.

## 2. Results and Discussion

### 2.1. General Considerations

The increasing number of sequenced plant genomes opens new horizons in the study of functional genomics and the precision of annotating new subfamily genes, manipulating them to potentially enhance plant performance and resilience to ever-changing environmental conditions. In this regard, AQP diversity has been studied in different ligneous species, including *Camellia sinensis* [42], *Citrus sinensis* [43], *Coffea canephora* [44], *Eucalyptus grandis* and *E. globulus* [45,46], *Gossypium hirsutum* [7,47], *Hevea brasiliensis* [48], *Jatropha curcas* [49], *Malus domestica* [50], *Olea europaea* [5], *Populus trichocarpa* [6,13], and *Vitis vinifera* [51]. Irrespective of whether AQP studies are carried out on perennial or annual species, they all demonstrate how AQPs support significant roles in the cellular homeostasis of different organs and tissues and in the radial and systemic transport of permeants, by increasing the permeability of various membranes to water and essential nutrients in response to any fluctuating demands (nutrition, transpiration, etc.) [10].

The AQP gene family exhibits a particularly high diversity and abundance in plants, most of which require further study for a better understanding. Current study on *Betula* AQPs has its own worth because, to the best of our knowledge, no study has been conducted on genome-wide identification, sequence characterization (i.e., diversity and structure), and the functional prediction of aquaporins in *Betulaceae*. Accordingly, a systematic genome-wide screening of the content of *AQP*-encoding genes was performed in the *B. pendula* genome. Sequence homology analysis and protein domain validation based on the analysis of conserved domains and protein topology resulted in the identification of 33 putatively functional and non-redundant AQP genes, named *Bpe*AQPs (Table 1). *Bpe*AQP full-length cDNAs and related amino acids and complete nomenclature are detailed in Appendix A.

### 2.2. Genome-Wide Identification, Diversity and Evolutionary Analysis of Betula Aquaporins

#### 2.2.1. Genetic Structure of *Bpe*AQP Subfamilies

The 33 *Bpe*APQs include members of each of the five major AQP subfamilies common to most other *Viridiplantae* (i.e., NIPs, PIPs, TIPs, SIPs, and XIPs) (Figure 1 and Appendix A). Ten proteins were grouped in the PIP and were differentiated according to the pattern of conserved motifs into two groups: PIP1 and PIP2; eight TIPs into five groups: TIP1, TIP2, TIP3, TIP4 and TIP5; eight NIPs into six groups: NIP1, NIP2, NIP4, NIP5, NIP6, and NIP7; four XIPs into two groups: XIP1 and XIP2; and three SIPs into two groups: SIP1 and SIP2 (Table 1 and Appendix A). Similar proportion and diversity of members were established in woody and annual species. It is worth mentioning that *Brassicaceae* and Monocots have no XIP subfamily [15]. Sequences belonging to hybrid intrinsic proteins (HIPs) and GlpF-like intrinsic proteins (GIPs) reported in the moss *Physcomitrella patens* were not found [12]. These present results are in agreement with those of earlier studies that showed that the number and diversity of AQPs in each subfamily are highly specific in plants.

When compared with other woody plants, the number of AQPs identified in *B. pendula* is similar to that in *J. curcas* (32 members [49]), *V. vinifera* (32 members [51]), or *C. sinensis* (34 members [43]), but significantly lower than that present in *M. domestica* (42 members [50]), *P. trichocarpa* (55 members [6]), *H. brasiliensis* (51 members, [48]), *G. hirsutum* (71 members [7]), or *O. europaea* (79 members [5]). The lower number of AQP genes encoded in the *B. pendula* genome could be explained in part by whole-genome duplication events different from those observed in the species where they are relatively higher [52,53,54]. However, the proximal localization of several homologs (<100 kb; [55]), their phylogenetic relatedness (>70% of similarity, [56]), and similar gene structures strongly allude to the occurrence of gene tandem duplication events in the evolutionary history of some regions of the *B. pendula* genome. It concerns two PIP2s (*BpePIP2;2* and *BpePIP2;3*) on chromosome 9 and the XIP subgroup on chromosome 3 and the contig 2937. The AQP gene families might have expanded through duplication [57,58], but, more generally, changes in the number of genes among the different species may be due to the size of their genome and/or to evolutionary processes for adaptation in the natural environment. Accordingly, gene duplication represents a main driving force in increasing genetic diversity, gene family size, and creating novel genes in Eukaryotes in general, and, in particular, in plants [59,60,61]. Our results for *Bpe*AQPs suggest that some PIP and XIP members might have expanded through duplication. Further investigations are therefore needed to clarify the evolutionary events that took place within the aquaporin family in the *Betula* genus, and whether the duplication of some members could be responsible for functional divergences within and between subfamilies.

#### 2.2.2. Genome Structure of *Bpe*AQP Genes Models

The 33 *Bpe*AQPs in our study could be assigned to twelve chromosome-forming groups (Appendix A). The chromosomal distribution varies greatly, from one member in four chromosomes (6, 7, 11, 19) to a high of 7 in chromosome 5. Chromosomes 5 and 8 contain three to five subfamilies of aquaporin genes, whereas other chromosomes carry one or two subfamilies of aquaporin genes. The chromosomal location of *Bpe*PIP1;1, *Bpe*TIP5;1, *Bpe*XIP2;1-like and *Bpe*XIP1;3, *Bpe*NIP1;2, and *Bpe*SIP1;2 could not be determined because of an incomplete physical map for *B. pendula*. This genomic distribution is consistent with previous studies showing that AQPs are unevenly distributed over a large number of chromosomes [5,44,50].

A comprehensive analysis of the structural properties of the genes, including exon/intron number and position, led to interesting conclusions regarding the possible origin, evolutionary relationship, and gene function among the different AQP isomers. As is commonly observed in multigenic families, such as plant AQPs, the exon–intron structure analysis detects the presence of a varying number and length of introns among the *Bpe*AQPs, contributing to significant variations in gene length (Figure 2b). The number of introns per *Bpe*AQP also varies from 1 to 4: the *Bpe*PIPs are characterized by three introns, the *Bpe*TIPs by two introns, the *Bpe*XIPs by two introns for the XIP2 clade and one intron for the XIP1 clade, the *Bpe*NIPs by a conserved four-intron structure (except for *Bpe*NIP5;2, which features three introns), and the majority of the *Bpe*SIPs display one to three introns. The fact that most members of the same subfamily share a similar exon/intron structure reinforces the observed phylogenetic distributions. Finally, *Bpe*PIP1;2, *Bpe*PIP1;4, and several members from the NIP and SIP subfamilies display the longest introns (>3 kbp).

Previous studies have established that the gain or loss of exons tied to natural selection are common features of the evolutionary process in plants genomes [62]. Perhaps most importantly, introns would interfere with the regulation of gene expression in different biological contexts, a phenomenon that is still under-estimated. Although most of them are spliced out during transcriptional or post-transcriptional maturations, introns can serve as important biological functions by elevating gene expression without functioning as a binding site for transcription factors; the phenomenon is called ‘intron-mediated enhancement’ [63,64]. The small number of differences in the gene structure displayed by some *Bpe*AQPs belonging to a same subfamily or between subfamilies suggest that these genes have evolved to perform different functions. Finally, the intron-exon organization of the *B. pendula* AQP subfamilies is similar to those in many other plants (*cf* references listed below), showing that AQPs are highly conserved in plants in an evolutionary way, whatever their taxonomic group.

#### 2.2.3. Expertizing Sequence Discrepancies for Few AQP Candidates

Among the *Bpe*AQPs, two PIP1 encoding partial aquaporin-like sequences, which are truncated and lack both of the two NPA motifs, were classified as pseudogenes and excluded from further sequence analysis (Appendix A).

Similarly, an XIP2 encoding partial aquaporin-like sequence (Bpev01.c1577.g0027) could also be classified as a pseudogene due to a truncated *N*-ter region of 51 amino acids (Appendix A). The genomes of *Betula plathyphylla* and *B. nana* display these two XIP2 versions in full-length and truncated versions (BPChr09G20962 and BPChr09G20935 in *B. plathyphylla*, respectively, and CAOK01002722 and CAOK01020514.1 in *B. nana*, respectively). However, an alignment of the two *Bpe*XIP2s with their XIP2 orthologs reveals that the *Bpe*XIP2;1 pseudogene sequence is 100% identical to the equivalent region in the full-length sequences of its related orthologs, whereas the *Bpe*XIP2;1 full-length version (Bpev01.c2937.g0004) has several divergent residues that are beyond the truncated region in question, but which are 100% identical to the equivalent region in the full-length sequences of its related orthologs (Appendix A). Rebuilding contigs with the equivalent truncated portion from Bpev01.c2937.g0004 with the pseudogene sequence (Bpev01.c1577.g0027) which occurs at the first intron generates a redefined theoretical protein which is 100% identical to its *B. platyphylla* and *B. nana* orthologs.

To reinforce our views on these sequence discrepancies, and the solution to solve them, it appears that transcriptional analyses by RNA-sequencing shows significant expression only for the pseudogene sequence (Bpev01.c1577.g0027), whereas the full original sequence (Bpev01.c2937.g0004) gives null or quasi-null expressions in all of the organs analyzed (*cf* part 2.4). Lastly, the alignments of the *B. pendula* whole transcriptome mRNA-sequencing reads cover and validate the existence of this new putative sequence. From our experience, the reason for the presence of this type of sequence in the database is usually due to an unvalidated identification method after genome sequencing and a *de novo* assembly of reads, which, in the case of *B. pendula*, was performed without a reference genome and experimental evidence. Therefore, this new theoretical full-length protein was included in all our analyses.

### 2.3. Protein Transporter Structure Analysis, and Conserved Substrate-Specific Residues of Betula AQP

The function of a protein is based on its 3D structure, determined by the primary protein sequence and amino acid interaction in the folded protein. For AQPs, the pore structure is key to understanding AQP function and examining the consequences of primary sequence variations to the 3D pore organization itself. A global 3D analysis is mandatory before any detailed scrutinization of the primary sequences.

#### 2.3.1. Protein and Pore 2D and 3D Basic Structure Organizations of *Bpe*AQPs

The solute specificity of the AQPs is determined by a series of amino acids whose orientation of the side chain lines the pores, creating various spatial constrictions. Although predictive, the conserved positions of selectivity motifs provide the key to understanding the basic organization of the pore, and these analyses can be strengthened by 2D and 3D analyses of the pore structure. In this regard, the homology-based tertiary protein structure and the biochemical properties of pore-lining residues (i.e., channel radius, length, polarity, charge, hydrophobicity, hydropathy, solubility, and ionizable residues) across the pore were predicted on individual *Bpe*AQPs protomers in the membranes by using Phyre2 and MOLE2.5 software tools (Table 2 and Appendix A).

The results confirm that each *Betula* AQP displays the six expected TM domains, which adopt an hourglass-like configuration and a transmembrane pore. Pore morphology analyses show significant diversity for the pore-lining residues which, de facto, affects regional hydrophobicity and charge across the tunnel. This diversity implies the occurrence of specificities between different *Bpe*AQPs to facilitate the permeation of different hydrophilic or hydrophobic solutes with different volume and chemical profiles.

Altogether, 21 *Bpe*AQPs show uniform and classical characteristics with hydrophilic pore openings towards both ends, and a more hydrophobic nature in the middle. Except for *Bpe*TIP3;1, which harbors significant enrichment of hydrophobic residues at both ends, the rest of the *Bpe*AQPs have hydrophilic residues at one end and a hydrophobic nature at the opposite end, where the hydrophilic end is then oriented either into the cytosol (*Bpe*PIP1; 4; *Bpe*XIP1;1, *Bpe*XIP2;1, *Bpe*NIP5;2, *Bpe*SIP2;1) within the vacuole (*Bpe*TIP2;1; *Bpe*TIP2;2, *Bpe*TIP4;1), or into the apoplasm (*Bpe*PIP2;2; *Bpe*PIP2;4; *Bpe*PIP2;5; *Bpe*XIP1;3).

Finally, every *Bpe*AQP member displays a hydrophilic zone along the channel, generally located at the most constricted areas that correlate with the NPA and/or ar/R motifs, depending on the member. Furthermore, the average pore diameters for the different subfamilies at the ar/R constriction range from 1.8 ± 0.22 Ångströms for the *Bpe*PIPs (i.e., 1.52 ± 0.11 Å and 2.2 ± 0.2 Å for the *Bpe*PIP2 and *Bpe*PIP1 groups, respectively), 1.92 ± 0.28 Å for the *Bpe*SIPs, 3.68 ± 0.55 Å for the *Bpe*NIPs, and 3.96 ± 0.31 Å for the *Bpe*XIP. The *Bpe*TIP subfamily exhibits an intermediate average pore diameter of 2.92 ± 0.44 Å.

The pore diameter averages of all the *Bpe*PIPs and of four *Bpe*TIPs are smaller than the ≈2.8 Å diameter of a water molecule. However, PIP and TIP protomers are able to form homo- and hetero-tetramer structures. These associations lead to intrinsic changes in each protomer structure, resulting in greater stability and protein folding which, in fine, enhances the performance of water transport across biological membranes [65]. These structural events concern the PIP1 protomers in particular which, originally, have no water permeability [66,67].

These molecular adjustments, which are functionally crucial for the global activity of the quaternary structure in water permeation of the *Bpe*AQP protomer diameters embedded in a tetramer configuration, need to be evaluated further, particularly for the *Bpe*PIPs and *Bpe*TIPs, by adopting appropriate computational models [68] involving functional and structural validations. Finally, the pore diameters of 5.2 Å for the *Bpe*NIP2 or the >3 Å for some of the *Bpe*TIPs, *Bpe*NIPs, and *Bpe*XIPs correlate with the diameters of the various predicted permeants described in this study (*cf* chapter 2.3.3), i.e., silicic acid and/or bulky hydrophobic solutes (e.g., glycerol, lactate, boric acid, and ammonia), respectively.

Overall, it would then be very interesting to specifically mutate some key pore-lining residues in order to validate the importance of several regulation points in the permeability control of *Bpe*AQPs.

Nevertheless, the purposes of asymetric electric charges remain an open research question. It is noteworthy that many AQPs are considered as “electrical dipoles”. The reasons for this electrical polarity are not fully understood yet, and still undergo investigation. It could be related to the electrical polarity of the biomembranes, therefore stabilizing the embedded protein. In addition, this contrast of charges could be key in the regulation of the intrinsic channel activities of the AQPs, knowing that these charges are quantitative and that the pH, phosphorylation, and Ca^2+^ could modulate them.

#### 2.3.2. Sequence Structure and Protein Function Relationship of *Bpe*AQPs

Plant AQPs present different functional characteristics driven by intrinsic biochemical properties [9,10]. The biochemical features of the 33 *Bpe*AQPs identified in our study were predicted, i.e., the protein size, molecular weight (MW), isoelectric point (*p*I), and the grand average of hydropathy (GRAVY) (Table 1). *Bpe*AQPs range from 231 to 321 (average 272) amino acids in length, molecular weights range from 24.49 to 32.84 (average 28.64) kD, and *p*I values range from 5 to 10.05 (average 7.59). The AQP subfamilies from different plant species classically share these features. Interestingly, the vacuolar proteins classically show a relatively lower *p*I (6.69) when compared to all other cytosolic proteins (*p*I 7.40) [69]. This lower average *p*I value of the *Bpe*TIPs (average 5.72), compared with the four other subfamilies (*p*I > 6.5, except for *Bpe*PIP2;2-3, *Bpe*XIP1s and *Bpe*NIP5;1 which have *p*I < 6.5), is linked to the absence of the basic residues in the *C*-terminal domains of the proteins. This relationship was previously described in *Arabidopsis* [2], eucalyptus [45] and olive [5]. The *p*I could reflect a functional constraint imposed on the MIPs, including the specific key regulation sites across the *C*-terminal regions involved in phosphorylation, methylation, sorting signals, and the interaction with regulating partners, which differ between MIP families. These results further demonstrate that aquaporins are involved in a versatile and dynamic regulation of solute movement [70]. Concerning the GRAVY parameters which reflect protein hydrophobicity and hydrophilicity, the scores are all positive (ranging from 0.314 to 0.913), indicating their hydrophobic nature, which, for an aquaporin, is a key property that facilitates high water and solutes permeability across membranes [17].

The putative conserved motifs among the *Betula* aquaporin members were detected by the MEME program. The results showed that the majority of the *Bpe*AQPs within the same group shared similar motifs, and that three of the motifs (motif 4 which corresponds to TM4, motif 8 to NPA2, and motif 3 to TM6) are highly conserved between all *Bpe*AQPs (Figure 2c and Appendix A). The other motifs are specific to one or several subfamilies, meaning that they may play a crucial role in particular functions. Moreover, the *Bpe*PIP subfamily shows the most motifs, whereas the *Bpe*SIP subfamily displays the fewest. These singularities are consistent with those of other plant species [5,44,52].

#### 2.3.3. Conserved Substrate-Specific Residue, and Solute Permeability of *Bpe*AQPs per Subfamily

Plant AQPs facilitate the transport of water and some small neutral molecules, such as glycerol, urea, boric acid, silicic acid, NH_3_, CO_2_, and H_2_O_2_. Multiple sequence alignment, coupled with atomic resolution and molecular dynamic simulations, reported residue compositions at key positions in the protein known to regulate AQP function, including dual NPA motifs, ar/R filter, FPs, and predictive SDPs. All these specific pore-lining residues contribute to determining which substrates would permeate through the AQP pore [71].

The conserved NPA motifs, which face each other on opposite sides, create an electrostatic repulsion of protons and act as a size barrier, contributing specifically to the selectivity of water molecules [72]. The ar/R filter, FPs, and predictive SDPs (SDP1–9) are essential for the selective transport of water and non-aqua substrate molecules [10,23,24].

Multiple sequence alignments reveal that NPA motifs are relatively well-conserved between subfamilies, whereas the ar/R filters and FPs are divergent and are characterized by high subfamily- and/or subgroup-specific residues (Table 1 and Appendix A).

##### PIP Subfamily

Multiple sequence alignments reveal that the dual NPA motifs are highly conserved in the *Bpe*PIPs and *Bpe*TIPs, showing the typical NPA residues of aquaporins (Appendix A). The *Bpe*PIP and *Bpe*TIP functions in water transport are very specialized, and their structures are strongly preserved in *Viridiplantae* [73], suggesting that they are subjected to strong selection pressure in most plant taxonomic lineages [74].

Some divergence occurs between the *Bpe*PIP members in the ar/R filter and FP signatures. The ar/R filters in all the *Bpe*PIPs are well-conserved and are composed of F, H, T, and R residues in TM2, TM5, LE1, and LE2, respectively. They are identical to the ar/R filters found in all plant PIPs; this hydrophilic motif is a signature of water and solute transporting aquaporins compared with the other subfamilies, which include more hydrophobic residues [21,22]. Additionally, FP P2–P5 in the *Bpe*PIPs exhibited identical amino acids (i.e., S, A, F, W); only the P1 position showed mixed residues (Q, G and M). This particularity has been reported in other plants such as olive [5], upland cotton [7], sweet orange [43], grapevine [51], rubber tree [48], jatropha [49] and cassava [52].

The sequence homology analysis of the *Bpe*PIPs with various orthologues shows that, depending on the PIP member, the *Bpe*PIP1s could be able to transport boric acid, H_2_O_2_, urea, and CO_2_, and the *Bpe*PIP2s could be able to transport H_2_O_2_ and urea (Appendix A) [24].

Some gases (e.g., CO_2_ and O_2_) diffuse through the membranes passively. However, several experimental data show that PIPs increase membrane permeability to CO_2_ in mesophyll and stomatal guard cells. With the noticeable exception of holoparasitic plants, CO_2_ is considered as a key substrate for plants (i.e., photosynthesis), and the facilitation of its membrane diffusion at the mesophyll seems to be exclusive of the plant PIPs [75,76]. More recently, the membrane diffusion of another gas, O_2_, was reported to be facilitated by *Nt*PIP1;3, and an increased *Nt*PIP1;3 transcript level was measured in *Nicotiana tabacum* roots after a seven-day hypoxia treatment [77], expanding the range of gas permeation possibilities provided by PIPs.

The conserved P2-P5 residues (A-F-W) were reported to be a signature for the CO_2_ transporter, which is shared by all *Bpe*PIPs. In this regard, SDP analysis predicts CO_2_ permeabilities for three *Bpe*PIP1s (*Bpe*PIP1;2, -3, -4), but not for *Bpe*PIP2s. Several PIP2s can mediate CO_2_ diffusion [78]. However, this feature would be made effective with PIP1-PIP2 tetramer configurations [74], and with the presence of an isoleucine (I) located at the end of the loop E [79]. This residue is present in five of the *Bpe*PIP2s (not for *Bpe*PIP2;4 where I is substituted by Leucine (L)) (Appendix A). However, SDP analysis does not predict CO_2_ diffusion for the *Bpe*PIP2s due to the replacement of I by a methionine (M) (SDP2) for all *Bpe*PIP2s, and some residue divergences in the SDP6 (D vs G-A-T) for three *Bpe*PIP2s (*Bpe*PIP2;1 -4 -6) (Appendix A). Most especially, it also reveals that three of the *Bpe*PIP2s (i.e., *Bpe*PIP2;2 -3 -5) show this single divergence on SDP1 (I vs M). To our knowledge, no experimental data has targeted this particular SDP2 residue. This observation might be related to the fact that mutations in the conserved amino acids in PIPs alter the capacity of the protein in the diffusion of CO_2_ [79]. All this together opens the possibility that CO_2_ can diffuse through some PIP2s in concert with certain PIP1s in *B. pendula*.

Our results show that, except for *Bpe*PIP2;2 and -3, all *Bpe*PIPs are predicted to facilitate the diffusion of H_2_O_2_. At low nanomolar levels, H_2_O_2_ acts as a central hub integrating the signaling network in response to biotic and abiotic stress and during reproduction and development processes [80]. Although it can diffuse passively through organelle and plasma membranes, AQPs were identified as the main candidate for the transport of H_2_O_2_; in this respect, some AQPs are now called peroxyporins [81]. Interestingly, H_2_O_2_ regulates the subcellular redistribution of peroxyporins [82] and their permeability by phosphorylation [83]. However, aside from the SDPs, the pore lining-residues involved in the selectivity and diffusion of H_2_O_2_ are not clearly characterized. Recently, growth assays of yeast cells expressing mutated *Hv*PIP2;5 have shown that Ser126 has a large impact on H_2_O_2_ transport with a minor influence on *Hv*PIP2;5-mediated water transport [84]. This serine is shared by all *Bpe*AQPs (Appendix A).

All the conserved domains observed in the *Bpe*PIPs support the idea that the PIP family is involved in various physiological reactions involving small molecules, which remain to be elucidated experimentally.

##### TIP Subfamily

Based on the ar/R filter homology, the *Bpe*TIPs are classified into four groups [85]. Group I is composed of *Bpe*TIP1;1, *Bpe*TIP1;2 and *Bpe*TIP1;3 (with ar/R residues: H, I, A, V), group IIa is composed of *Bpe*TIP2;1 and *Bpe*TIP2;2 (H, I, G, R), group IIb is formed with *Bpe*TIP3;1 and *Bpe*TIP4;1 (H, I, A, R), and group III includes only one member, *Bpe*TIP5;1 (N,V,G,C) (Table 1 and Appendix A). This last group is relatively rare in most plant species [86]; and our results confirmed the variability of the TIPs observed in plants. The diversity observed in the ar/R selectivity filter in the different *Bpe*TIP subgroups is similar to the TIPs from other plant species [5,6,7,13,42,43,44,45,46,47,48,49,50,51]. The FPs are well conserved between members, mainly composed of (T, S, A, Y, W), except for *Bpe*TIP5;1 with (V, A, A, Y, W) residues. The TIPs, conjointly with the PIPs, are considered to be the main channels controlling water balance in plants. However, when compared to the *Bpe*PIPs, the *Bpe*TIPs exhibit a greater divergence in the residues of the ar/R selectivity filter and FPs. Consequently, based on the ar/R filter, the *Bpe*TIP-I and -II groups have a wider pore aperture, which might facilitate the permeation of relatively larger substrates than water, such as H_2_O_2_, urea, and ammonia [7,87].

Depending on the *Bpe*TIP member, the conserved residues ensure the transport of various solutes across membranes (Table 1) [24]. For example, TIPs are able to transport nitrogenous compounds [88], such as urea or ammonia [87,89]. Ammonia transport is a singularity of the TIPs in plants. It is dependent on H2 and H5 positions (H and I residues, respectively) with a non-polar LE1 (A/G) [90] and on the R vs V substitution at the LE2 position [91]. Interestingly, a histidine (H), localized in the loop C and only observed in *Bpe*TIP2s and *Bpe*TIP4;1, would be key for the NH_4_^+^ deprotonation [92].

One member per TIP group is predicted to transport H_2_O_2_ (Table 1) [24]. The transport of these solutes has been proven experimentally in heterologous systems [87,89,93,94] and mutagenic studies [95]. Our results suggest that *Betula* TIPs from groups I and II are able to transport these solutes across membranes (Table 1). Even though the role of H_2_O_2_ in the vacuole is unclear, it is thought to be mostly associated with ROS detoxification and the activities of heme-containing and flavin-dependent oxidoreductases, Cu/Zn SOD, or type III peroxidases (PRX) that are able to generate/utilize H_2_O_2_ to oxidize a wide range of metabolites.

To close, *Bpe*TIP5;1 is the only aquaporin in group III. Its ortholog, *At*TIP5;1, shares similar FPs, and can transport urea [87]. It is therefore plausible that *Bpe*TIPs provides the same transport.

##### XIP Subfamily

The *Bpe*XIP family is composed of four members (and two pseudogenes), and has a lower overall sequence identity compared with other AQP subfamilies. The three *Bpe*XIP1s belong to the XIP-A group, and the XIP2;1 belongs to the XIP-B clade [14]. Compared with other subfamilies, little is known about this recently discovered family, probably due to its absence in the most extensively studied model plants, including several dicot families, such as *Brassicaceae* (i.e., *A. thaliana*), or the entire monocot clade with *T. aestivum*, *O. sativa* and *Z. mays*, for example.

Every *Bpe*XIP harbors conserved (N/S)P(I/L)-NPA motifs which are usually found for this XIP subfamily in other plant species (Appendix A). Several variations in ar/R selective filter for XIP family members have been reported in different studies [6,12]. *Bpe*XIPs display H2, H5, and LE1 amino acids with polar uncharged (Threonine) or hydrophobic side chains (Valine, Isoleucine, Alanine), increasing the hydrophobicity of the constriction. The hydrophobic nature of XIPs facilitates transport of bulky and hydrophobic molecules, such as glycerol, urea, and boric acid in plants [96].

However, it is still unknown what kind of functional effect this residue change might have on the protein, and no transport function has been described for the XIP-A subfamily, although they are predicted to transport urea or glycerol [6,24].

Regarding the XIP-B subfamily, *Bpe*XIP2;1 is predicted to transport H_2_O_2_ and urea [21], which is supported by experimental studies proving that *Nt*XIP1;1 [92] or *Hb*XIP2;1 [11] facilitate the diffusion of these solutes with glycerol. Besides that, *Nt*XIP1;1 overexpression in *N. tabacum* results in disturbed boron tissue distribution, leading to boron deficient phenotypes in meristems and young leaves [97].

Froger’s positions are globally conserved; however, no clear role has been attributed to them in the transport selectivity of the XIP protein.

A singularity concerns *Poptr*XIP2;1, which is the only XIP known to be capable of transporting small amounts of water (for information, *Nt*XIP1;1 [97] and *Hb*XIP2;1 [11] are non-permeable to water). The truncated version of the *Bpe*XIP2;1 shares similar dual NPAs and ar/R SF signatures with *Poptr*XIP2;1 (as well as with the XIP2 truncated sequences from *B. nana* and *B platyphylla*). These conserved dual NPA motifs appear to be rare [15] and are observed with *Poptr*XIP2;1 from *P. trichocarpa* [14]. Therefore, like *Poptr*XIP2;1, it is plausible that a complete version of the *Bpe*XIP2;1 copy which displays these dual NPAs is able to transport small amounts of water. However, these sequences are amputated of their first exon (which corresponds to the *N*-ter extension), making them silent. The fact that a plausible but “rare” water-permeable version of XIP can exist in a species, although becoming non-functional, is intriguing. As such, this evolutionary occurrence is very interesting and needs to be proven experimentally.

##### NIP Subfamily

Concerning the NIP subfamily, the two NPAs show the same sequence as the PIPs and TIPs, except for the *Bpe*NIP5s and *Bpe*NIP6;1 where the alanine is replaced by a serine residue in the first NPA, and by a valine in the second NPA motif (Appendix A). The divergences concern the ar/R filter and the FPs, suggesting various substrate transport selectivity for these subfamily members and, putatively, the involvement in differential physiological roles.

The NIP family forms a monophyletic group, mainly predicted to localize in the plasma membrane [98]. It belongs to the aquaglyceroporins, a subset of the AQP family, and displays more diverse substrate specificities than the PIP and TIP subfamilies, playing a major role in the transport of glycerol, lactic acid, urea, and many metalloids, including arsenic, boric and silicic acids, and antimony [99,100].

Eight NIPs were identified in the *Betula* genome. They were classified into the three NIP subgroups based on their ar/R filter, suggesting that each subgroup of the NIPs has its own function. The group I is composed of *Bpe*NIP1;1, *Bpe*NIP1;2 and *Bpe*NIP4;1; the group II is composed of *Bpe*NIP5;1, *Bpe*NIP5;2, *Bpe*NIP6;1, and *Bpe*NIP7;1; and the group III includes the *Bpe*NIP2s. The *Bpe*NIP ar/R filters were almost identical to those of *C. sinensis* [43] and *O. europaea* [5], where these genes are predicted to act as water facilitators [85].

The *Bpe*NIPs belonging to the group I harbor the classical (W-V-A-R) in H2, H5, LE1, and LE2 residues that would be more hydrophobic than those observed for the group II members which harbor (A-I-G/A-R), (S-I-A-R) or (A-V-G-R) residues. These divergences in residues would impact the transport ability of the protein, suggesting that each subgroup of the NIPs has its own function. For example, although the typical (W-V-A-R) residues are directly involved in the water transport, it was reported that this signature confers a low water permeability compared with the other groups, while enhancing the transport of uncharged solutes, such as glycerol and formamide [101]. Furthermore, the NIPs that combine the ar/R signature (A-I-G/A-R) with the dual NPS/NPV motifs are able to transport arsenite, boric acid, and silicic acid in rice [102,103].

In addition, group II is very interesting in many ways. The *Bpe*NIP5s are very close to *At*NIP5;1, which seem to be capable of transporting arsenite, boric acid, and antimony in experimental assays [100,104]. The *Bpe*NIP6;1 is phylogenetically close to *At*NIP6;1, which is able to transport urea and the amino acid glycine (Gly) in addition to metalloids [105,106]. As for the *Bpe*NIP7;1, its phylogenetic divergence does not separate it from group II and its proximity to *At*NIP7;1 suggests that it can perform similar functions by transporting arsenite, boric acid, antimony, urea, and Gly [104]. Interestingly, the presence of a specific tyrosine residue in the TM2 (Y81) of *At*NIP7;1 has been directly related to gating and regulation of urea and Gly transport [107]. The *Bpe*NIP7;1 displays this polar and aromatic residue in TM2 (Y91) (Appendix A), and the ability of the *Bpe*NIP7;1 to transport these permeants must be considered.

The NIP-group III is present in the *Betula* genome, and the NIP2 members compose it. Their ar/R selectivity filter is unique and is composed of the typical G, S, G, and R residue (Table 1 and Appendix A). The small size of the amino acid in the H2 and H5 positions, and their precise distance of 108 amino acids distance to the NPA domains, generate the largest pore diameter (>4.38 Å) described in aquaporins, allowing the passage of very large solutes such as silicic acid, arsenite, and boric acid when expressed in *Xenopus* oocytes [16]. Mutations in the H5 position result in a loss of transport activity for these molecules [101]. In the plant kingdom, different species accumulate varying degrees of silicon (Si) [108,109]; this group III is present in several silicon-accumulating plants [105]. In addition, NIP2 members from *Oryza sativa* or *Zea mays* are able to transport arsenite, boric acid, antimony, urea, H_2_O_2_, and Gly (Table 1 and Appendix A) [21,100,110,111,112]. *Bpe*NIP2;1 is phylogenetically assigned to the same clade as the *Populus trichocarpa* and *Solanum lycopersicon* NIP2s (Appendix A) (as well as *O. sativa* and *Z. mays* NIP2s, data not shown), suggesting that the *Bpe*NIP2;1 could be viewed as a Si transporter. The plant species considered as high accumulators of Si are known to accumulate up to 10% of Si on a dry weight basis [113], whereas the species with a low Si-accumulating capacity (i.e., around 0.2% or less of Si) lack NIP2s or functional NIPs without a precise distance between the NPA domains and the ar/R selectivity filter. Among the angiosperms, *Fagales*, which includes *B. pendula*, presents significant shoot Si concentrations [114], suggesting that *Betula* can be a potential silicon-accumulating species. However, to confirm the spatial predispositions of the NPA and the ar/R selectivity filter of the *Bpe*NIP2;1 and its Si-permeability, experimental validation is required.

##### SIP Subfamily

The SIPs are divided into two groups based on sequence alignments and on both the ar/R selectivity filter and FPs. In general, the NPA domains of the SIPs are highly divergent from conventional AQPs, especially in the first NPA domain [115]. These features are found in the *Bpe*SIPs (Table 1 and Appendix A); the first NPA motif shows the replacement of alanine by threonine (T; *Bpe*SIP1;1 and PvSIP1;2), serine (S, *Bpe*SIP1;2) or leucine (L; *Bpe*SIP2;1), while the second NPA motif is completely conserved in the other subfamilies. The ar/R positions from SIP aquaporins show valine/alanine/serine in H2, threonine/lysine in H5, proline/glycine in LE1, and asparagine/serine in LE2. All these residue variations are classically shared by the SIP subfamily in other plant species.

Even today, the SIP protein basic structure, solute specificity, and function *in planta* are still under characterization. Water channel activity was determined for the *At*SIP1s, unlike for *At*SIP2;1; they are subcellularly localized to the endoplasmic reticulum [116]. However, no data on transport of non-aqueous substrates is available. The variety in the ar/R selectivity filters and FPs, in comparison with those of other MIPs, suggests that the water channel function may not be the sole function of the SIPs. In this regard, homologues- and structure-based analyses predict that the SIP1s could be capable of transporting urea, while the SIP2s could transport H_2_O_2_ and arsenite [117].

Finally, although every global substrate specificity study, mainly based on prediction, has to be carefully considered, our phylogenetic analysis links the *Bpe*SIPs to their orthologues in *A. thaliana*, *P. trichocarpa*, *L. esculentum*, and *O. europaea* (Appendix A). Even though the *Bpe*SIPs show slight residue variations at the selectivity filters, an attribution of solute transport to these proteins can be predicted.

#### 2.3.4. Subcellular Localization Prediction of *Bpe*AQPs

Predicted spatial features are helpful for the functional characterization of AQPs. Every *Bpe*AQP member exhibits a highly conserved tridimensional structure composed by six transmembrane helices (Table 1 and Appendix A). The highly conserved hydrophobic regions with specific amino acid sequence and the TM number support the transmembrane structural integrity of the AQPs, and possibly for PIP and TIP protomers. The spatial orientation of specific highly conserved residues is key in the formation of a stable tetramer [118].

The plant aquaporins are mainly located in the plasma membrane. However, specific membrane localizations can vary between the different AQP subfamilies, which, ultimately, influence sub-cellular flow and compartmentalization of solutes. Almost all of the *Bpe*AQPs are predicted to be localized to the plasma membrane, except for the *Bpe*TIP5;1 which would be exclusively assigned to the chloroplast (Table 1 and Appendix A). The *Bpe*PIPs are all predominantly assigned to plasma membranes and, very marginally, to tonoplasts, or to ER and Golgi endomembranes. As for the other *Bpe*AQP subfamilies, subcellular localizations are contrasted to varying degrees between members from a same subfamily, being assigned to the vacuole, endoplasmic reticulum, Golgi complex, peroxisomes, mitochondria, and chloroplasts.

Different PIP and TIP members were detected in mitochondrial and chloroplast membranes [119,120], where they could transport key solutes (CO_2_, water and H_2_O_2_, nitrogen) to various plant metabolisms [121,122]. A part of the predictions we obtained in this study is in agreement with the data reported in the literature, but some differences were also observed. In addition, some predicted cell compartmentalizations of several *Bpe*AQPs seem “unusual”, such as the TIP and SIP in chloroplast membranes, or the NIP and XIP in tonoplasts. No location should be excluded without further experimental demonstrations, and more studies are required to verify these preliminary data.

### 2.4. Profiling of BpeAQP Transcriptome

Plants are constantly exposed to a large variety of abiotic stresses. Cold stress (CS, chilling, 0–15 °C; freezing, <0 °C) has a strong impact on plant growth and development, and directly influences the species expansion, crop distribution, and yield [123].

Cold stress tolerance in plants is a very complex trait. Studies on chilling and freezing tolerance from woody species, including silver birch, have shown that CS directly influences various physiological and metabolic reactions which are regulated by profound changes in gene expression [124,125]. However, the involvement of AQPs in the acclimation and/or tolerance to chilling is poorly understood. To provide insight into the physiological roles of the various *B. pendula* AQP isoforms, publicly available whole-transcriptome RNA-seq datasets [33] were processed to study the early expression patterns of the 33 *BpeAQP* genes in leaves under a cold stress of 6 °C (Figure 3).

The early responses of the *BpeAQPs* to CS show that the strongest transcriptional modulations are limited to a few isoforms and mainly vary at 1 and 1.5 h after stress application (Figure 3). However, although several trends in the trimmed mean of M-values (TMM) can be distinguished, they are not statistically significant (*p* > 0.05) (Appendix A), mainly due to the paucity of biological repetitions (i.e., two or three biological replicates depending on the kinetic point), which calls for the repetition of the experiment.

The major trends are an up-regulated group with *BpePIP2;5*, *BpePIP2;1*, *BpeNIP6;1* and *BpeSIP2;1*, and a down-regulated one with *BpePIP1;2, BpePIP1;3*, *BpePIP2;4*, *BpeNIP1;2*, and *BpeTIP1;1*. The early responses of various plants to cold stress are reported in the literature, and they concern transcriptomic analysis and transgenic expression [126]. Authors have reported differential expression patterns with down- or up-regulations according to the AQP members (including several *Bpe*AQP orthologues), organs, and genotypes. Interestingly, some *BpeAQP* genes show similar transcription accumulation kinetic with orthologs from *O. europaea* (*OleurPIP1;2*, *OleurPIP1;3*, *OleurPIP1;1*, *OleurPIP2;5* [5]), *Musa acuminate* (*MaPIP2;4*, *MaPIP2;5* [127]), rice (*OsPIP1;2*, *OsPIP2;5*, *OsTIP1;1* [128]) or *A. thaliana* (*AtPIP1;2*, *AtPIP1;3*, *AtPIP2;4*, *AtPIP2;5* [129]) during cold stress. These data suggest that a set of specific aquaporin genes respond to cold acclimation in the leaf of birch, and that particular MIP sequences (e.g., PIP1;2, PIP2;5, TIP1;1) shared by different plant species plausibly play key roles for adaptation to cold stress in plants. In addition, transgenic plants overexpressing some PIPs or TIPs displayed an enhanced tolerance to cold [130,131]. In our results, we could not detect the involvement of the few modulated *BpeAQP* genes in the CS stress; however the down co-modulation of the *BpePIP1s* and *BpePIP2;4*, for example, suggest plausible hetero-tetramerization events between protomers, which would minimize water loss, or the up-regulation of two *Bpe*PIP2s which may facilitate CO_2_ transport from leaves under cold stress.

In spite of these trends, we are not able to propose definitive functional interpretations of the modulation of the *Bpe*AQP isoforms. However, it is becoming clear that the role of the AQPs in the regulation of physiological status under hydraulic constraint is highly complex and still largely unraveled, especially for the plant species that share important multigenic families [132]. In this regard, the two highly expressed *Bpe*PIP2s in leaves are the *BpePIP2;1* and the *BpePIP2;4* (Figure 4), which harbor up- and down-regulated expression during CS, respectively. The opposing patterns suggest compensating events between isoforms, where one AQP group supplements the functions of another group. This compensation could create a differential flow of permeants and possibly a preferential orientation of their flow, revealing a complex participation of different aquaporins in matter flow between cytosol and the different subcellular compartments and/or apoplasm [68].

It is particularly noteworthy that low temperatures have a very discrete impact on only a few isoforms in *B. pendula*. Several other studies showed substantial and large AQP modulations during cold stress, such as olive [5] or banana [127]. In contrast to these Mediterranean or tropical species that are extremely cold-sensitive, *B. pendula* is a rather cold-resistant species. In this regard, the *Betula* genus is represented by a pioneer species which occupy a broad latitudinal range in the Northern Hemisphere [133], from the sub-tropics to the Arctic, and populating various contrasted habitats, ranging from bogs, highlands, to alpine tundra, swampy mountain habitats, and forests with more or less drained soil. Therefore, in view of the prime functions that aquaporins play in the regulation of water and solute homeostasis under diverse environmental conditions, it becomes legitimate to ask the following question: do the discrete modulations in the early expression patterns of *BpeAQP* genes correlate with the high resilience of *Betula* leaf to CS, and, more widely, with the high plasticity of this species? It will be very interesting to study the involvement of AQPs in plants growing in very contrasting environments and under various abiotic stresses, under the prism of the phenotypic plasticity of *B. pendula*.

We also compared the expression patterns of the 33 *BpeAQP* genes in different organs, including flowers, leaves, roots, and xylem (Figure 4).

The comparison of normalized gene expression levels shows that 23 *BpeAQPs* were expressed in at least one of the organs, 11 of which showed significant variations regarding organ type (Figure 4a and Appendix A). Overall, the *Betula* AQP gene family displays complex differential expression profiles according to the organs, including 18 members that show ubiquitous expression, whereas 11 members exhibit organ-specific expression patterns, thereby suggesting different physiological functions in the organs during developmental, and mineral and/or water absorption processes. Furthermore, out of all the *Bpe*AQPs, the levels of expression observed for the PIP and TIP subfamilies are higher than that for the NIP subfamilies. In addition, transcript accumulations are greater in the root than all of the other organs, xylem being the least represented in this abundance (Figure 4b). These expression profiles are in agreement with previous studies in other plant species such as *Brassica rapa* [55], *Oryza sativa* [58], and *Eucalyptus* [46].

The main aquaporin expressed in the different organs is *BpeTIP1;1*, predominantly in the leaves (1 253 TMM, Appendix A) and in the flowers (1 165 TMM, Appendix A), as compared with the average of the other *BpeAQPs* at 250 TMM (Appendix A). *BpeTIP1;2* seems to be specific to roots (888 TMM, Appendix A), but the highest *BpeAQPs* expressed in this organ is *BpePIP2;5*. The TIPs ensure high water transport capacity, especially with the TIP1s, reportedly responsible for the high permeability of the tonoplasts [134].

The fact that only one isoform (i.e., *BpeTIP1;1*) among the TIPs is highly expressed in all of the organs, and that some other *BpeAQPs* exhibit organ-specific expressions (e.g., *BpeTIP1;2, BpePIP1;3, BpePIP2;5* and *BpePIP2;6* in the roots; *BpeNIP1;2* and *BpePIP1;2* in the flowers; *BpeTIP1;1* and *BpePIP2;4* in the leaves and flowers; *BpePIP2;1* in xylem) could be due to a redundancy of functions between counterparts, or to the specialization in particular organs and/or tissues, as revealed with *BpeTIP2;2*, which is the only isoform that would be able to facilitate the diffusion of NH_4_^+^ (Table 1 and Appendix A) [24]. Moreover, the TIPs in general and the TIP1s in particular are highly involved in the cell division and elongation processes [135,136], suggesting that the *Bpe*TIP1;1 could be mainly responsible for the water transport between the cytoplasm and vacuole in *B. pendula* plants. The *BpeTIP2;2*, which expresses in the roots, suggests an ammonium transport facilitation and compartmentation into the vacuole, protecting cells from ammonium toxicity [137].

The functions carried out by the *Bpe*TIP1;2 and *Bpe*TIP2;2 in the root require further elucidation. In our analysis, the *BpeTIP3;1* gene showed no expression in any of the organs studied, even in the flowers for which the TIP3 subfamily is known to be specifically and/or strongly expressed [138]. These results suggest that the TIP3 function might not be strictly conserved between plant species, as has also been reported with olive [5], *J. curcas* [49], banana [127], and carnation [139]. We hypothesize that other AQP proteins, such as the *Bpe*TIP1;1, which shows high expression in the flowers and in the leaves, may be a functional substitute in these organs. Although the aquaporins are quite functionally redundant overall (Table 1), these results support the notion that the functions of the same subgroup can be differentiated between organs, as well as between different species.

The PIP subfamily has the largest number of expressed isoforms, including all *Bpe*PIP1s and five out of the six *Bpe*PIP2s. Every *BpePIP1* expresses in all of the organs tested, but contrasted accumulations can be highlighted with *BpePIP1;2*, which is expressed in xylem as well as in the leaves and flowers, whereas *BpePIP1;3* predominantly accumulates in the flowers and roots. The involvement of PIP1s in the gene network associated with leaf, root, and xylem responses is relatively well-documented, although more so under drought, salinity, and heat conditions. However, there is less study on the involvement of PIP1s with the flower responses, where *PIP1;2* showed higher expression in some plant species, and where the related protein was shown to be involved during the flower opening stages, in water transport, and petal expansion [139,140,141]. Regarding the *Bpe*PIP2s, *BpePIP2;1* mainly accumulates in xylem, *BpePIP2;4* in the flowers and leaves, and *BpePIP2;5* and *BpePIP2;6* in the roots. Interestingly, *BpePIP2;3* shows very weak expression in the roots and xylem (Appendix A), whereas *BpePIP2;2* shows no expression in any organ. These two genes are duplicated genes. However, although the duplicate sequences did not diverge significantly during their evolution (*K*s/*K*a < 1), and where purifying selection might have contributed greatly to the maintenance of the function in common, each duplicate appears to have undergone different fates, including one of the copies which has become silenced over time (i.e., non-functionalization). Gene duplication events are considered as a primary driving force, providing raw material for evolutionary novelty. In the case of *BpePIP2;2* and *-3*, the duplication does not appear to have occurred with the accumulation of mutations in both copies, meaning that the creation of the duplicate should not create functional redundancy, which would affect the fitness of the plant. However, overlapping functions between duplicate genes can result in interfering interactions between paralogues, including major repercussions on related steady-state mRNA and/or protein pools and, potentially, on the regulatory mechanisms controlling various physiological processes with far-reaching phenotypic effects [142,143]. Clearly, the exact functional contribution of each *Bpe*PIP2 paralogue will need to be established.

The more weakly expressed subfamilies (TMM < 250, Appendix A) are the NIP with *BpeNIP1;2*, *BpeNIP5;1* and *BpeNIP6;1*; SIP (where the three members show expression with TMM averaging 50); and the XIP with the *BpeXIP2;1* mainly (TMM ≈ 50). These low expression levels are characteristically observed in other species. However, it is likely that their expression might change in response to a specific stimulus or that they are expressed at higher levels but in very specific cell types making up a small population of the total organ sampled and analyzed for RNA-seq.

Overall, RNA expression is a very informative first step for assessing the involvement of the particular members in a biological context, and additional proteomic efforts are needed to confirm these differential AQP abundances in various organs.

## 3. Material and Methods

### 3.1. Identification of BpeAQP Family Members and Subfamily Classification

*BpeAQP* genes were searched against the sequenced birch genome available on NCBI (https://blast.ncbi.nlm.nih.gov/Blast.cgi; accessed on 2 July 2021) and the CoGe comparative genomics platform (https://genomevolution.org/coge/). The investigations were conducted using keyword queries (“Major Intrinsic Protein” and “Aquaporin”), complemented by tBLASTn searches with conservative criteria requiring a cut-off of *E*-value of 1.0^−5^ by using the amino acid sequences of aquaporin sequences from *A. thaliana* (*At*AQP) [2] and *Olea europaea* L. (*Olea*AQP) [5] as queries. The *B. pendula* AQP sequences (*Bpe*AQP) were named according to the standardized MIP nomenclature, and each name (i.e., PIP, TIP, NIP, SIP, and XIP) was guided by sequence similarity and phylogenetic analysis with orthologs from *A. thaliana*, *O. europaea, Lycopersicon esculentum* (*Sl*AQP; [144]) and *Populus trichocarpa* (*Poptr*AQP, [14]). Each *BpeAQP* gene accession is derived from the identifier of the GenBank GSS contig on which it is located. When necessary, it is subscripted with an additional lowercase letter to indicate that *BpeAQP* genes accessions differing only by this letter were originally assembled in the same GSS contig. The *Betula* AQP protein names, accession numbers, and sequences (genomic, CDS, and deduced proteins) used in this work are listed in Figure 1, Appendix A, respectively.

### 3.2. Bioinformatics Analysis

Every putative *Bpe*AQP sequence was carefully scrutinized, including expected AQP motifs (NPA, ar/R selectivity filter, and the FPs) and the prediction of the transmembrane topology, with Interproscan from EMBL (http://www.ebi.ac.uk/Tools/pfa/iprscan/; accessed on 2 July 2021) and the NCBI’s conserved domain database (https://www.ncbi.nlm.nih.gov/Structure/cdd/wrpsb.cgi; accessed on 2 July 2021). Motifs were identified by multiple sequence alignment and performed using MUSCLE (https://www.ebi.ac.uk/Tools/msa/muscle/; accessed on 2 July 2021) [145] with the default parameters. Specificity-determining positions (SDP1–9; [21]), which were essential to the transport of non-aqua substrates, were identified from multiple sequence alignments as described by [146]. Molecular weight, theoretical isoelectric point, and the grand average of hydropathicity (GRAVY) of the *Bpe*AQPs were analyzed by the ExPaSy compute pI/Mw tool (https://web.expasy.org/protparam/; accessed on 2 July 2021). The transmembrane regions were predicted using the TMHMM-2.0 software tool (www.cbs.dtu.dk/services/TMHMM/; accessed on 2 July 2021) [147], SOSUI tools (http://harrier.nagahama-i-bio.ac.jp/sosui/sosui_submit.html; accessed on 2 July 2021) [148], and were, when necessary, manually corrected with heterologous *At*AQP, *Sl*AQP and *Olea*AQP sequences. The subcellular localizations of the *Bpe*AQPs were predicted using the online tools WoLF PSORT (http://wolf-psort.hgc.jp; accessed on 2 July 2021) and Plant-mPLoc (http://www.csbio.sjtu.edu.cn/bioinf/plant-multi/; accessed on 2 July 2021) [149]. The phylogenetic studies were conducted with the *Bpe*AQP-deduced peptide sequences, and then with a total of 221 amino acid sequences, including *Bpe*AQPs, *At*AQPs, *Olea*AQPs, *Poptr*AQPs, and *Sl*AQPs. Alignments were performed using the ClustalW progressive alignment method, and the phylogenetic trees were inferred using the maximum parsimony method. Maximum parsimony analyses were conducted using the subtree-pruning-regrafting (SPR) algorithm, and were bootstrapped with 5000 replicates. All analyses were performed using the MEGA X program. The graphical representation and the edition of the phylogenetic trees were performed with iTOL software (https://itol.embl.de/; accessed on 2 July 2021) [150]. The intron and exon structures of *BpeAQP* genes were analyzed by online software GSDS 2.0 (http://gsds.gao-lab.org/; accessed on 2 July 2021) [151]. Conserved motifs of the *Bpe*AQP proteins were identified by MEME Suite 4.11.1 (Multiple Expectation Maximization for Motif Elicitation; http://meme-suite.org/tools/meme; accessed on 2 July 2021) [144]. The motif discovery mode was selected as a classic mode by using the default settings, i.e., the zero or one occurrence per sequence (ZOOPS) site distribution per sequence, of a minimum width of 6, a maximum width of 50 amino acid motifs, and a maximum number of motifs to find which was set to 10.

### 3.3. Construction of Homology-Based Tertiary Protein Structure

Homology-based protein tertiary structures (PDB files) of all the *Betula* AQPs were predicted using the Phyre2 protein-modeling server (http://www.sbg.bio.ic.ac.uk/phyre2/html/page.cgi?id=index; accessed on 2 July 2021) [152], using the intensive modelling mode as parameter. The results obtained in the form of PDB files were then uploaded to the Mole2.5 server to predict the transmembrane pores and various biochemical properties of the pore-lining residues (https://mole.upol.cz/; accessed on 2 July 2021) [153]. The following default parameters were also used: heteroatoms were ignored for modelization and pore merging was set on. Channel modelization was obtained with cavity probe radius, 5 Å; cavity interior threshold, 1.1 Å; channel origin radius, 5 Å; channel surface cover radius, 5 Å; channel weight function, the Vororoi scale; bottleneck radius, 1.2 Å; bottleneck tolerance, 3 Å; and maximum tunnel similarity, 0.7 Å.

### 3.4. Differential Expression Profile of BpeAQP Gene Family (RNA-Seq Analysis)

The transcriptional expression patterns of all of the *BpeAQPs* from various plant organs and, subsequently, in leaves undergoing cold stress were recorded from the plant samples that were previously studied in two experimental assays on *B. pendula* [33]. The transcriptomic data are available in the NCBI SRA (Sequence Read Archive) database with accession numbers PRJNA535361 and PRJNA532995, respectively.

Samples were collected in triplicate from the roots, young leaves, female inflorescences, and xylem (upper stem, about 10th nodes) of healthy two-year-old birch planted in the experimental field of the Northeast Forestry University (Harbin, China). Two biological replicates per sample were sequenced, and each biological replicate consisted of a pool of three plant RNAs. The Northeast Forestry University experimental field is situated at 45.72° north latitude and 126.63° east longitude. The average daylight is 14.25 h and the nighttime is 9.75 h, the mean temperature is 17 °C to 27 °C, and the average relative humidity about was 76% at the date of sampling.

Concerning the cold stress assay, samples were taken from young leaves from two-month-old *B. pendula* plants grown in the greenhouse at a constant temperature of 25 °C with a photoperiod of 16 h of light and 8 h of dark. A total of six 28W sunlamps were used for illumination, but the intensity of illumination was not measured specifically. For the experimentation, the plants were exposed to cold stress (6 °C) for 0.5 h, 1 h, 1.5 h, 2 h, 2.5 h, and 3 h. Plants left at 25 °C were used as the control (0 h). Two biological replicates for time points of 1 h, 2 h, 2.5 h, and 3 h, control plants, and three biological replicates for time points of 0.5 h and 1.5 h were generated.

Total RNA was extracted using the CTAB (Cetyltrimethylammonium Bromide) method [154]. The constructed cDNA libraries were subjected to paired-end sequencing using the Illumina HiSeq platform at Biomarker Technologies Corporation (Beijing, China). The clean reads of each sample were obtained by filtering out any reads of low quality, and then aligned to the *B. pendula* reference genome using Bowtie2 [155]. The RNA-sequencing data were analyzed using the RNA-seq by Expectation-Maximization (RSEM) pipeline [156]. RSEM could compute transcript abundance, estimating the number of RNA-seq fragments corresponding to each gene, and normalized expression values as TMM (trimmed mean of M-values). Every methodological step is detailed in [33].

Statistical analyses were performed with R (version 3.6.3). The following key packages were used: bestNormalize, version 1.6.1 [157]; broom, version 0.7.4 [158]; FactoMiner, version 2.4 [159]; factoextra, version 1.0.7 [160]; ggplot2, version 3.3.3 [161]; janitor, version 2.1.0 [162]; multcomp, version 1.4-16 [163]; pander, version 0.6.3 [164]; rstatix, version 0.6.0 [165]; and tidyverse, version 1.3.0 [166].

AQPs with zero expression values were considered as non-available data, when distribution analysis showed non-normal distribution of such zero values. Normality and homoscedasticity were checked before analysis of variance (tested at 5% risk level) (Appendix A). A post-hoc Tukey Honestly Significant Difference (HSD) test was performed at 5% risk level and a Student’s *T* test was performed at 5% risk level with Benjamini and Hochberg FDR adjustments for all AQPs *p-*values [167]. Principal component analysis was carried out on the first two dimensions and calculated using the first five dimensional variables as parameterized by default in FactoMineR package. The models of variance considered cold stress (CS), time duration of the cold stress (6 °C), and the organ type for the samples analyzed in the different libraries of the organs sampled at 25 °C.

## 4. Conclusions

Species of the *Betula* genus occupy large natural landscapes, playing a key role in forestry and horticulture. Profound climate changes increase the vulnerability of various woody species, making aquaporins (as being vital regulators of the plant water relationship) relevant candidates in developing stress-resistant forest plants. The present experiment identifies that the *B. pendula* genome harbors 33 genes encoding putative AQPs. Our results determined the variability of the AQPs in *Betula*, allowing us to establish five *Bpe*AQP subfamilies. The evaluation of the *Bpe*AQP sequences reveals a great diversity in the NPA motifs, ar/R selectivity filters, and FPs, which might promote the transportation of water and a large panel of unconventional non-aqua substrates throughout the plants. This exhaustive study on the *Bpe*AQP specificity to different substrates is only based on predictions. Functional tests in heterologous or homologous systems need to be performed to analyze the functional role of specific *Bpe*AQPs in more detail. Nevertheless, the diverse transport characteristics of *Bpe*AQPs are highly informative and potentially indicate their various roles in the key physiological processes involved in development and low-temperature tolerance in *Betula*, thus opening up new horizons for very interesting research.

## Figures and Tables

**Figure 1 ijms-22-07269-f001:**
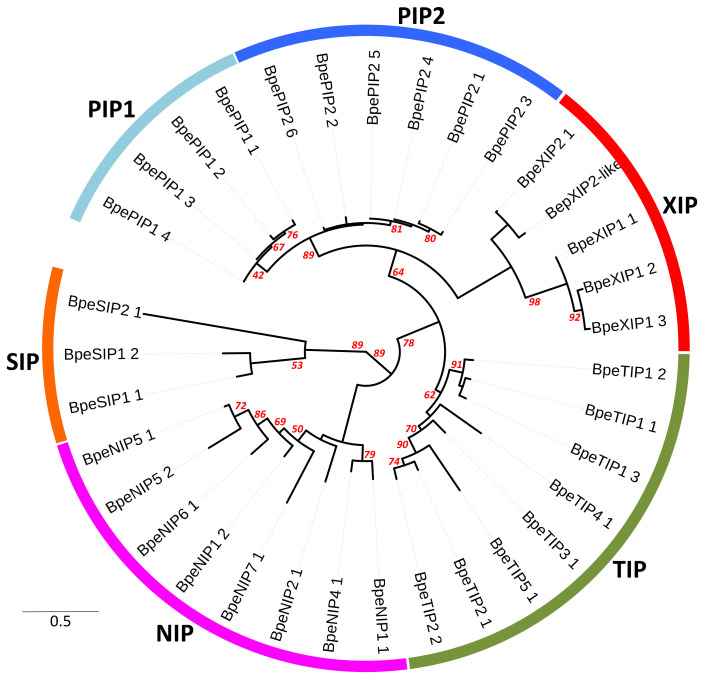
Phylogenetic analysis of the *Bpe*AQP full-length and the truncated *Bpe*XIP2;1-like protein sequences from *Betula pendula*. Deduced amino acid sequences were aligned using ClustalW, and the phylogenetic tree was constructed using the maximum parsimony method. Maximum parsimony analysis was conducted using the subtree-pruning-regrafting algorithm. The number next to the branch’s nodes represents bootstrap values ≥50% based on 5 000 resamples. The distance scale denotes the number of amino acid substitutions per site. The name of each subfamily is indicated next to the corresponding group. The *Bpe*AQP accession numbers and sequences are listed in Appendix A. The complete phylogenetic analysis of aquaporin family proteins of *Betula pendula* (*Bpe*AQPs, filled circles) with the AQP sequences from *Arabidopsis thaliana* (*At*AQPs), *Lycopersicon esculentum* (*Sl*AQPs), *Olea europaea* (*Oeu*AQPs) and *Populus trichocarpa* (*Poptr*AQPs) is proposed in Appendix A.

**Figure 2 ijms-22-07269-f002:**
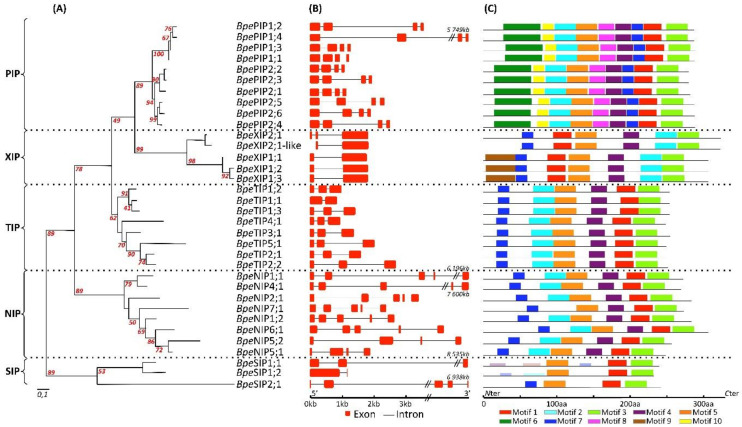
(**A**) Phylogenetic relationship, (**B**) Exon/Intron genomic structure, and (**C**) Protein motif organization of the 33 full-length and the pseudogene *Bpe*XIP2;1-like *Betula pendula* aquaporin sequences. (**A**) Deduced amino acid sequences were aligned using ClustalW, and the phylogenetic tree was constructed using the maximum parsimony method. Maximum parsimony analysis was conducted using the subtree-pruning-regrafting algorithm. The number next to the branch’s nodes represents bootstrap values ≥ 50% based on 5000 resamples. *Bpe*AQPs clustered into five AQP subfamilies: *Bpe*PIPs, *Bpe*XIPs, *Bpe*TIPs, *Bpe*NIPS and *Bpe*SIPs. (**B**) Exons and introns of the *BpeAQP* genes are represented by red boxes and black lines, respectively. Gene structures were compared using GSDS software. Gene orientations are indicated (5′–3′) in the *x*-axis. (**C**) Distribution of the conserved motifs among the *Bpe*AQP proteins. Motif analysis was performed by using the MEME web server. Ten conserved motifs were identified, and the different motifs are identified using different colored boxes, as indicated at the bottom of the Figure. Each color block in the different proteins indicates a specific motif, for which the amino acids are detailed in Appendix A. Protein orientations are indicated (*N*ter-*C*ter) on the *x*-axis.

**Figure 3 ijms-22-07269-f003:**
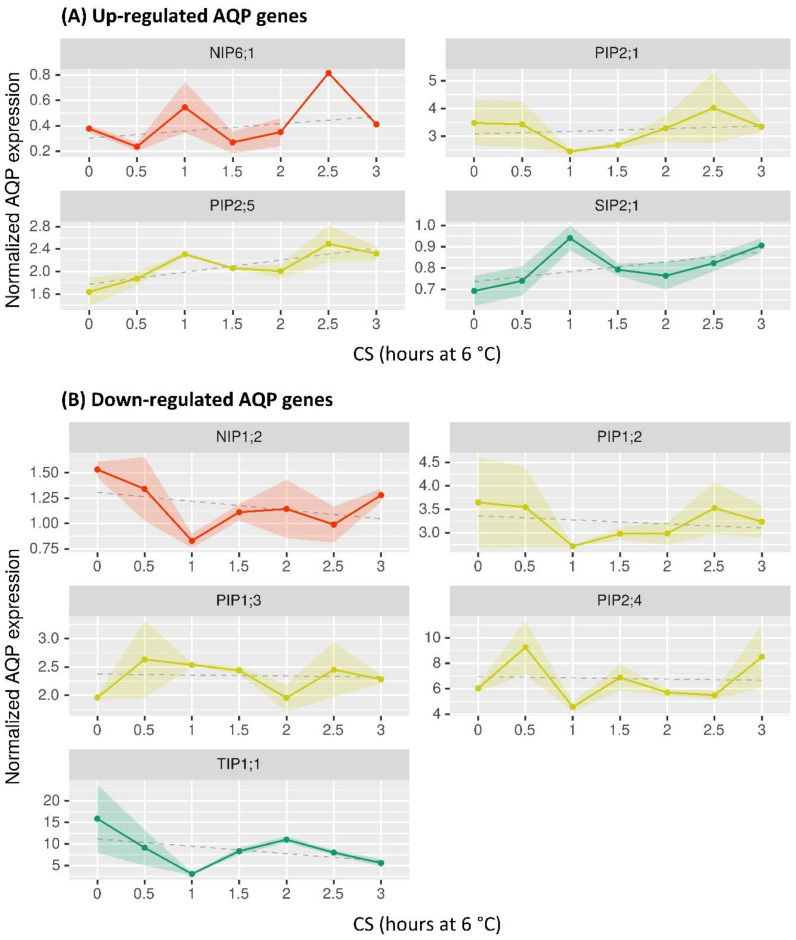
Early transcriptional expression profiles of the *BpeAQP* genes from *Betula pendula* leaves in response to cold stress (6 °C). The *BpeAQPs* are color-coded by subfamily. Data are represented by normalized expression level as a function of CS at 6 °C duration in hours. (**A**) Group of up-regulated AQPs in leaves; (**B**) Group of down-regulated AQPs. The grouping is based on the linear trend (dashed line) estimated by modelization using all time points. PIP, plasma intrinsic protein; TIP, tonoplast intrinsic protein; NIP, nodulin-like intrinsic protein; and SIP, small basic intrinsic protein.

**Figure 4 ijms-22-07269-f004:**
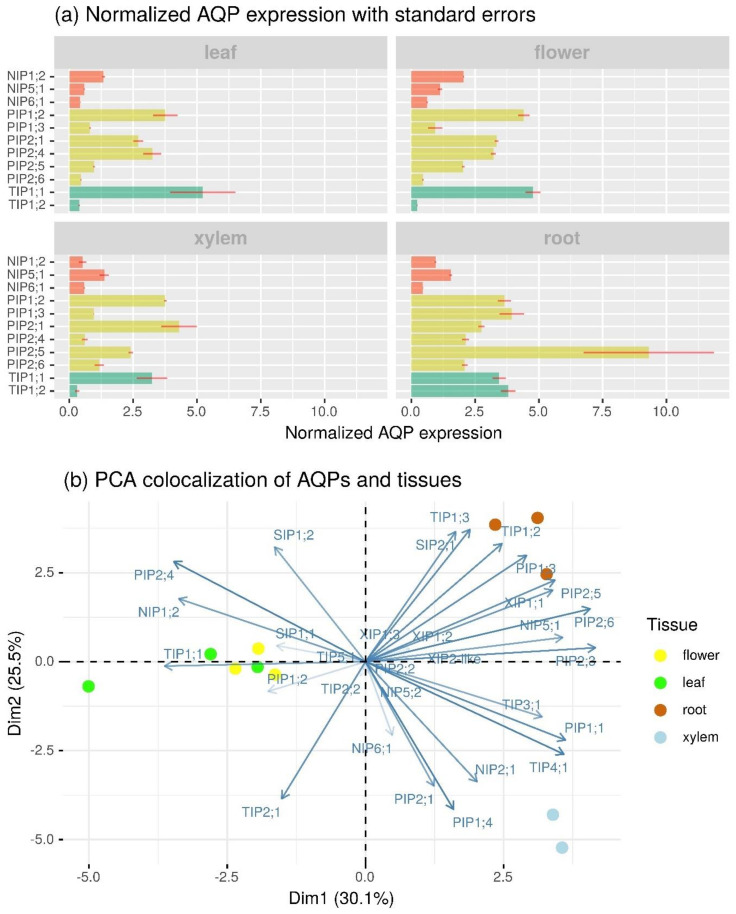
Organ-specific expression profiles of the *BpeAQP* genes from *Betula pendula*. (**a**) Normalized AQPs expression per organ type. Error bars are based on standard error of the mean. The *Bpe*AQPs are color-coded by subfamily. (**b**) PCA biplot based on the *Bpe*AQP expression levels and organ types. The organs are color-coded as per the legend indicated on the graph. The plane represents 55.6% of the total variance analyzed. PIP, plasma intrinsic protein; TIP, tonoplast intrinsic protein; XIP, *X*-intrinsic protein; NIP, nodulin-like intrinsic protein; SIP, small basic intrinsic protein.

**Table 1 ijms-22-07269-t001:** Nomenclature and protein properties of the *Bpe*AQPs from *Betula pendula* (complete sequences).

^a^ Loci	Size	^b^ MW	^b^*p*I	^b^ GRAVY	^c^ TMH	^d^ SubCL	^e^ NPA		^f^ ar/R SF	^g^ Froger’s	^h^ Predicted Transport
Proposed Gene Name/Locus	(aa)	(kDa)					LB	LE		Residues	Substrate
**Plasma membrane Intrinsic Proteins (PIPs)**											
***Bpe*****PIP1;1**/Bpev01.c0190.g0072.m0001	287	30.86	9.26	0.314	6 (5) *	PM	NPA	NPA	F-H-T-R	Q-S-A-F-W	Boron H_2_O_2_ Urea
***Bpe*****PIP1;2**/Bpev01.c0170.g0038.m0001	286	30.51	8.76	0.417	6	PM-Vac	NPA	NPA	F-H-T-R	G-S-A-F-W	Boron H_2_O_2_ Urea CO2
***Bpe*****PIP1;3**/Bpev01.c1699.g0001.m0001	288	30.89	8.61	0.384	6	PM-Vac	NPA	NPA	F-H-T-R	Q-S-A-F-W	Boron H_2_O_2_ Urea CO2
***Bpe*****PIP1;4**/Bpev01.c0027.g0083.m0001	286	30.69	9	0.357	6 (5) *	PM-Vac	NPA	NPA	F-H-T-R	Q-S-A-F-W	Boron H_2_O_2_ Urea CO2
***Bpe*****PIP2;1**/Bpev01.c0658.g0014.m0001	281	29.79	8.83	0.48	6	PM-Vac	NPA	NPA	F-H-T-R	M-S-A-F-W	H_2_O_2_ Urea
***Bpe*****PIP2;2**/Bpev01.c0577.g0022.m0001	278	29.86	6.72	0.527	6	PM-Vac	NPA	NPA	F-H-T-R	M-S-V-F-W	Urea
***Bpe*****PIP2;3**/Bpev01.c0577.g0023.m0001	278	29.91	6.2	0.44	6	PM-Vac	NPA	NPA	F-H-T-R	M-S-A-F-W	Urea
***Bpe*****PIP2;4**/Bpev01.c0042.g0019.m0001	287	30.41	8.68	0.498	6	PM-Vac-ER	NPA	NPA	F-H-T-R	N-S-A-F-W	H_2_O_2_ Urea
***Bpe*****PIP2;5**/Bpev01.c0552.g0003.m0001	286	30.79	8.25	0.424	6	PM	NPA	NPA	F-H-T-R	Q-S-A-F-W	H_2_O_2_ Urea
***Bpe*****PIP2;6**/Bpev01.c0483.g0002.m0001	287	30.44	8.93	0.506	6	PM-Vac-Gol	NPA	NPA	F-H-T-R	Q-S-A-F-W	H_2_O_2_ Urea
**Tonoplast Intrinsic Proteins (TIPs)**											
***Bpe*****TIP1;1**/Bpev01.c0278.g0002.m0001	252	26.39	5.8	0.624	6	PM-Vac	NPA	NPA	H-I-A-V	T-S-A-Y-W	H_2_O_2_ Urea
***Bpe*****TIP1;2**/Bpev01.c0396.g0015.m0001	252	25.81	4.95	0.843	6	Vac-PM	NPA	NPA	H-I-A-V	T-S-A-Y-W	Urea
***Bpe*****TIP1;3**/Bpev01.c2330.g0012.m0001	252	26.15	5	0.755	6	PM-Vac	NPA	NPA	H-I-A-V	T-S-A-Y-W	Urea
***Bpe*****TIP2;1**/Bpev01.c0665.g0011.m0001	247	25.19	6.15	0.866	6	PM-Vac	NPA	NPA	H-I-G-R	T-S-A-F-W	H_2_O_2_ Urea
***Bpe*****TIP2;2**/Bpev01.c0120.g0064.m0001	250	25.44	5.11	0.913	6	Vac-PM	NPA	NPA	H-I-G-R	T-S-A-Y-W	H_2_O_2_ Ammonia Urea
***Bpe*****TIP3;1**/Bpev01.c1026.g0003.m0001	253	27.11	6.7	0.874	6	PM-Vac-ER	NPA	NPA	H-I-A-R	T-A-A-Y-W	H_2_O_2_ Urea
***Bpe*****TIP4;1**/Bpev01.c0921.g0003.m0001	247	25.91	5.72	0.774	6 (7) *	Vac-PM	NPA	NPA	H-I-A-R	T-S-A-Y-W	Urea
***Bpe*****TIP5;1**/Bpev01.c0477.g0022.m0001	248	25.24	6.39	0.75	6	Chlo-Vac	NPA	NPA	N-V-G-C	V-A-A-Y-W	H_2_O_2_ Urea
**Uncharacterized *X* Intrinsic Proteins (XIPs)**											
*Bpe*XIP2;1 Redefined sequence	321	30.08	8.35	0.623	6	PM-Vac-ER	NP*V*	NPA	I-T-A-R	V-C-P-F-W	H_2_O_2_ Urea
***Bpe*****XIP1;1**/Bpev01.c1577.g0026.m0001	305	32.84	5.67	0.724	6	PM-ER-Chlo-Vac-Pero	NP*I*	NPA	V-I-V-R	V-C-P-F-W	Urea
***Bpe*****XIP1;2**/Bpev01.c1577.g0028.m0001	305	32.77	6.29	0.764	6	PM-Vac-ER-Gol	NP*I*	NPA	V-I-V-R	V-C-P-L-W	-
***Bpe*****XIP1;3**/Bpev01.c2937.g0004.m0001	305	32.76	6.06	0.704	6	PM-ER-Vac-Chlo	NP*I*	NPA	V-I-V-R	V-C-P-F-W	-
**Nodulin-26 like Intrinsic Proteins (NIPs)**											
***Bpe*****NIP1;1**/Bpev01.c0145.g0009.m0001	270	28.47	8.33	0.715	6	PM-Vac	NPA	NPA	W-V-A-R	F-S-A-Y-I	Ammonia Urea
***Bpe*****NIP1;2**/Bpev01.c0230.g0002.m0001	282	29.73	9.25	0.425	6	PM-Golg-ER-Vac	NPA	NPA	W-V-A-R	F-S-A-Y-L	Ammonia Urea
***Bpe*****NIP2;1**/Bpev01.c0281.g0063.m0001	282	29.5	8.43	0.405	6	PM-Vac-ER	NPA	NPA	G-S-G-R	L-T-A-Y-L	Boron Urea
***Bpe*****NIP4;1**/Bpev01.c1162.g0002.m0001	269	28.57	9.05	0.67	6	PM-ER	NPA	NPA	W-V-A-R	F-S-A-Y-I	Ammonia Urea
***Bpe*****NIP5;1**/Bpev01.c1045.g0004.m0001	247	25.94	8.91	0.737	6 (5) *	Vac-PM-Mito-ER	NP*S*	NP*V*	A-I-G-R	F-T-A-Y-L	Urea
***Bpe*****NIP5;2**/Bpev01.c1084.g0010.m0001	255	26.39	5.39	0.938	6 (5) *	PM-Vac	NP*S*	NP*V*	A-I-A-R	F-T-A-Y-M	Boron Urea
***Bpe*****NIP6;1**/Bpev01.c0044.g0051.m0001	305	31.46	8.51	0.41	6	PM-Vac-ER-Gol	NPA	NP*V*	S-I-A-R	F-T-A-Y-L	Boron Urea
***Bpe*****NIP7;1**/Bpev01.c0330.g0006.m0001	272	29.12	8.13	0.508	6 (5) *	PM-ER-Vac-Gol	NPA	NPA	A-V-G-R	Y-S-A-Y-M	-
**Small basic Intrinsic Proteins (SIPs)**											
***Bpe*****SIP1;1**/Bpev01.c0082.g0002.m0001	239	25.34	9.76	0.781	6	Vac-PM-ER-Gol-Chlo	NP*T*	NPA	V-T-P-N	F-A-A-Y-W	-
***Bpe*****SIP1;2**/Bpev01.c0387.g0011.m0001	231	24.49	10.05	0.775	6	PM-Chlo-Gol-Vac	NP*S*	NPA	A-T-P-N	F-A-A-Y-W	-
***Bpe*****SIP2;1**/Bpev01.c0212.g0010.m0001	240	26.27	9.4	0.626	6 (4) *	Vac-PM-ER	NP*L*	NPA	S-K-G-S	I-V-A-Y-W	-

^a^ Loci, Gene IDs and AQP location are based on CoGe assembly v1.0. ^b^ MW, Protein molecular weight; *p*I, protein isoelectric point; GRAVY, Grand Average of Hydropathy. ^c^ TMH, Number of transmembrane helices predicted by TMHMM and SOSUI analysis tools; * regions were adjusted by alignments with characterized orthologs from *Arabidopsis*, Poplar and Tomato. ^d^ SubCL, Predicted subcellular localization by WoLF PSORT and Plant-mPLoc analysis tools; the first mention corresponds to the most statistically preponderant subcellular localization. PM, plasma membrane; Vac, Vacuole; ER, Endoplasmic reticulum; Gol, Golgi; Chlo, Chloroplaste; Mit, Mitochondria; Per, Peroxisome. ^e^ NPA, Asparagine, Proline, Alanine; Bold italic letters denote unusual amino acids in the NPA motifs. ^f^ ar/R SF, ar/R selectivity filters (H2-H5-LE1-LE2). ^g^ Froger’s residues (P1-P2-P3-P4-P5). ^h^ Potential substrate transported prediction using the signature sequences developed by [21]. The five aquaporin subfamilies are highlighted in the shaded rows.

**Table 2 ijms-22-07269-t002:** Physicochemical properties of the pore structure predicted for the protein of *Bpe*AQPs from *Betula pendula* (complete squences).

Aquaporins	^a^ size (X—Y—Z)	^b^ Channel	^b^ ar/R Bottleneck	^b^ Bottleneck	^c^ Hydropathy	^d^ Charge	^e^ Polarity	^f^ Mutability	Lipophilicity		Solubility	^j^ Ionizable
	(Å)	Length (Å)	Radius (Å)	Radius (Å)					^g^ *LogP*	^h^ *LogD*	^i^ *LogS*	
**Plasma membrane Intrinsic Proteins (PIPs)**												
***Bpe*** **PIP1;1**	41.358—46.156—64.972	50.3	1.2	1.1	0.7	0	9.08	86	0.61	0.41	−0.25	2
***Bpe*** **PIP1;2**	75.341—52.171—52.769	52.7	0.8	0.8	1.45	2	6.45	85	1.04	0.86	−0.6	2
***Bpe*** **PIP1;3**	41.338—49.248—63.209	49	1.2	1.1	1.01	−1	6.75	87	0.71	0.55	−0.29	3
***Bpe*** **PIP1;4**	79.903—52.414—52.769	40.5	1.2	0.6	1.46	1	5.41	83	0.83	0.74	−0.53	1
***Bpe*** **PIP2;1**	77.283—50.592—50.381	51.3	0.7	0.5	1.38	2	6.56	90	0.92	0.76	−0.47	2
***Bpe*** **PIP2;2**	73.807—57.060—51.485	52.2	0.7	0.6	1.07	0	9.75	87	0.94	0.69	−0.49	2
***Bpe*** **PIP2;3**	73.807—57.060—51.485	52.4	0.9	0.5	0.84	1	9.59	87	0.83	0.57	−0.31	3
***Bpe*** **PIP2;4**	76.112—54.225—51.485	48.6	0.7	0.6	0.99	0	10.36	88	0.92	0.66	−0.45	2
***Bpe*** **PIP2;5**	76.894—50.699—50.542	47.4	0.9	0.6	0.78	−1	12.73	87	0.94	0.58	−0.42	3
***Bpe*** **PIP2;6**	79.375—57.747—50.542	48.9	0.7	0.5	1.04	1	11.23	88	1.03	0.74	−0.58	3
**Tonoplast Intrinsic Proteins (TIPs)**												
***Bpe*** **TIP1;1**	72.798—51.488—50.744	43.2	1.3	0.5	0.93	−1	8.19	91	0.83	0.66	−0.44	1
***Bpe*** **TIP1;2**	72.546—44.617—51.466	60.5	1.3	0.4	0.92	0	6.22	84	0.88	0.7	−0.35	2
***Bpe*** **TIP1;3**	40.145—37.773—54.582	58.4	0.9	0.6	1.14	0	4.49	91	0.67	0.67	−0.37	1
***Bpe*** **TIP2;1**	40.061—37.320—52.603	54.3	1.7	0.4	0.5	1	9.59	88	0.53	0.41	−0.32	1
***Bpe*** **TIP2;2**	38.810—37.320—50.132	40	1.6	0.6	0.98	1	7.58	84	0.61	0.5	−0.4	1
***Bpe*** **TIP3;1**	41.075—36.911—53.559	52.9	1.7	0.6	1.18	1	6.4	85	0.84	0.7	−0.54	3
***Bpe*** **TIP4;1**	41.339—38.620—53.616	60.6	1.5	0.6	0.89	1	9.55	84	0.8	0.63	−0.5	1
***Bpe*** **TIP5;1**	72.604—38.620—53.617	43.5	2.3	0.5	0.3	1	5.27	88	0.37	0.23	−0.01	1
**Uncharacterized *X* Intrinsic Proteins (XIPs)**												
***Bpe*** **XIP2;1**	46.806—47.507—65.542	59.3	1.7	1.1	0.99	0	6.89	85	0.74	0.56	−0.34	2
***Bpe*** **XIP1;1**	77.536—59.089—66.029	57.8	1.8	0.7	0.95	2	5.72	87	0.46	0.27	−0.13	2
***Bpe*** **XIP1;2**	40.311—46.757—67.465	48.8	2	1.3	1.1	2	6.23	88	0.74	0.52	−0.29	2
***Bpe*** **XIP1;3**	76.853—56.997—54.194	48	2.4	0.4	1.16	1	8.73	86	0.88	0.63	−0.27	3
**Nodulin-26 like Intrinsic Proteins (NIPs)**												
***Bpe*** **NIP1;1**	47.170—47.961—60.868	45.4	1.2	0.7	0.94	1	3.99	91	0.64	0.52	−0.29	1
***Bpe*** **NIP1;2**	48.080—48.913—64.173	34.8	1.4	1.2	0.41	1	6.97	93	0.28	0.09	0.15	1
***Bpe*** **NIP2;1**	45.851—47.139—66.412	51.6	2.6	0.8	0.51	2	7.02	92	0.47	0.31	0.09	2
***Bpe*** **NIP4;1**	50.644—47.660—62.545	52	1.2	0.8	0.89	2	6.13	85	0.71	0.56	−0.35	2
***Bpe*** **NIP5;1**	46.077—36.449—63.012	55.5	2.1	0.7	0.13	0	8.43	86	0.6	0.4	−0.22	2
***Bpe*** **NIP5;2**	45.644—34.894—57.891	52.5	2	0.9	0.5	0	9.26	90	0.63	0.32	−0.08	2
***Bpe*** **NIP6;1**	46.270—46.230—62.913	45.2	1.7	0.8	0.61	0	7.6	85	0.47	0.32	−0.11	2
***Bpe*** **NIP7;1**	48.706—48.254—67.609	60.8	2.5	1	0.16	0	12.76	82	0.52	0.26	−0.01	2
**Small basic Intrinsic Proteins (SIPs)**												
***Bpe*** **SIP1;1**	76.118—45.635—48.786	48.4	1,2	0.9	0.22	0	2.03	91	0.33	0.33	−0.07	N/A
***Bpe*** **SIP1;2**	41.601—38.523—52.141	46.8	0.7	0.6	−0.11	3	8.57	87	0.53	0.2	−0.03	3
***Bpe*** **SIP2;1**	76.963—48.686—44.305	51.3	0.7	0.5	0.37	2	8.27	84	0.47	0.2	0.06	4

^a^ size (X-Y-Z), sizes calculated using Phyre2. They include the cytosolic *C*- and *N*-terminals of the proteins. ^b^ Physicochemical properties calculated by using MOLE2.5. ^c^ Hydropathy, average of hydropathy index per each aminoacid according to Kyte and Doolittle (1982). Range from from the most hydrophylic (Arg = −4.5) to the most hydrophobic (Ile = 4.5). ^d^ Charge, charged amino acid residues (positive Arg, Lys, His; negative Asp, Glu). ^e^ Polarity, Average of lining amino acd polarities. Polarities range from nonpolar (Ala, Gly = 0), through polar (e.g., Ser = 1.67) to charged (Glu = 49.90 and Arg =52.00). ^f^ Mutability, Average of relative mutability, based on empirical substitution matrices between similar protein sequences. ^g,h^ Lipophilicity, logP and logD-scales, Octanol/water partition coefficients of channel-surrounding fragments, calculated via www.chemicalize.org. ^i^ Solubility, logS-scale, Water solubility of channel-surrounding fragments, calculated via www.chemicalize.org. ^j^ Ionizable, Ionizable residues. The five aquaporin subfamilies are highlighted in the shaded rows.

## Data Availability

All data are contained within the article and Appendix A.

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
