# Peer review of "Genome-Wide Identification, Structure Characterization, and Expression Pattern Profiling of the Aquaporin Gene Family in *Betula pendula"

_ijms, 2021, doi:10.3390/ijms22147269_

Round 1

Reviewer 1 Report

Dear Authors,

The manuscript regarding the Genome-wide study of the aquaporin gene family in Betula pendula is well written and performed. l. However, the manuscript raises a minor question. Otherwise, I will recommend for accepted after minor revision

Authors have conducted Cold stress to check the impact on plant growth, but in the abstract section (in lines 34-36), they said that AQP genes have functional role in water availability. The author can select waterlogging or dehydration stress instead of cold stress for this study. Please modify the abstract accordingly.

In Bioinformatics analysis section 3.2. The authors did not mention the parameters of bioinformatics tools. Please include the parameter of analysis where possible.

Author Response

Dear Authors,

The manuscript regarding the Genome-wide study of the aquaporin gene family in Betula pendula is well written and performed. l. However, the manuscript raises a minor question. Otherwise, I will recommend for accepted after minor revision

[Authors] Thank you for your positive remarks and comments on our work. All our responses are highlighted in red.

Authors have conducted Cold stress to check the impact on plant growth, but in the abstract section (in lines 34-36), they said that AQP genes have functional role in water availability. The author can select waterlogging or dehydration stress instead of cold stress for this study. Please modify the abstract accordingly.

[Authors] It is true that AQPs play key roles in the water flow that maintains the general metabolism and cellular homeostasis. However, in the context of our study, and specifically for this sentence, we have to conclude with "cold stress" instead of "water availability". This sentence has been rewritten accordingly (lines 36-37).

Furthermore, in order to be fully in line with this modification, we have added a clarification in the introduction that highlights the link between cold stress and aquaporins. (lines 122-129)

In Bioinformatics analysis section 3.2. The authors did not mention the parameters of bioinformatics tools. Please include the parameter of analysis where possible.

[Authors] These bioinformatics tools are well adapted to the study of transmembrane proteins, and our analyses were undertaken mostly using the default parameters. However, parametrizations of bioinformatic tools were added correspondingly in the text where needed as requested (that concerns Plant-mPLoc, MEME and MOLE2.5 tools). (lines 914-917, 921, 924-928). However, it is noteworthy that Plant-mPloc and Wolf-PSORT propose no default parametrization since this is automatically inferred by the algorithm from data proposed by the user. For Phyre2, we used the automatic parametrization mode “Modeling mode” set as “intensive”; thus detailed parametrization is inferred as well from given data.

Reviewer 2 Report

Definitely, here is an interesting paper with large volume of data and supplementary material. However, the paper still requires essential revisions before being considered further.

1) Quality of figure S8 is low, it’s not readable.

2) Figure S12, it’s reasonable to give the full name in the figure legend for TMM values, not the abbreviation initially.

3) Table S1, FXXK01000578.1a , FXXK01000578.1b , FXXK01001578.1b , FXXK01001578.1a , FXXK01001578.1c , FXXK01002938.1a , FXXK01002938.1b are not found in ncbi. Pls. check and correct, the nucleotides without letters are OK.

4) Pls, read and consider to cite + more papers from the book:

Gambetta, G.A.; Knipfer, T.; Fricke, W.; McElrone, A.J. Aquaporins and root water uptake. In Plant Aquaporins. Signalling and Communication in Plants; Chaumont, F., Tyerman, S., Eds.; Springer: Cham, Germany, 2017; pp. 133–153. 

5) Pls, refer to a few first papers on aquaporins dated back to1980s-1990s with functional characterization (e.g. in frog oocytes), the citation list is skewed for recent papers of the last years.

6) a variety of inorganic and organic solutes such as inter 95 alia boron, silicon, ammonia, glycerol, hydrogen peroxide, urea, polyols, glycine

Silicon and boron are transported in the forms of acids,not the mentioned elements.

7) The Introduction is basically good and comprehensive.

8) are under 117 investigation [32-34] where aquaporins are vital regulators of plant-water relations. 118

The role of aquaporins is actually overestimated. The main points for plant water relations are the water potentials. Pls, have a look at the mentioned paper at point 4.

9) The genomical, structural and biochemical features were determined on 123 the entire set of BpeAQPs, therefore classifying each sequence in the general plant AQP 124 system.

Not determined but predicted based on the present software, the data were not obtained/confirmed experimentally.

10) Table 1, not readable well. Pls, correct. Especially, the first, last and TMH columns.

11) Figure 1.

The numbers are not provided for all the pairs. Pls, add or discuss.

12) Same as 11) for figure 2.

13) Figure 1, there are 34, not 33 aquaporins depicted at the picture. Pls, check all and correct somewhere (text or figure) with XIP2-like (preferably before the figures 1, 2).

14) Table 2. the first two columns are not readable. Pls, correct.

15) need to be evaluated further, particularly for the BpePIPs and 359 BpeTIPs, by adopting appropriate computational models [62].

Need crystal structure and experimental results, not computer models.

16) With the noticeable exception of holoparasitic plants, CO2 435 is considered as an universal substrate for plants, and the facilitation of its membrane 436 diffusion at the mesophyll seems to be exclusive of the plant PIPs

Not clear about universal substrate, too philosophical. Pls, rephrase in a clear form.

17) All this together opens the possibility that some 455 PIP2s could diffuse CO2 in concert with certain PIP1s in B. pendula.

Not correct phrase, PIP2 can not diffuse, CO2 can diffuse.

18) The specific membrane localization can vary between the different AQP subfamilies, 636 which, ultimately, influences sub-cellular flow and compartmentalization of solutes. Al-637 most all of the BpeAQPs are predicted to be localized to the plasma membrane, except for

Not correct phrase, see your own text earlier for localization and its description.

19) However, although several trends in trimmed mean of M-675 values (TMM) can be distinguished, they are not statistically significant (p>0.05) (Figure 676 S12) mainly due to the paucity of biological repetitions, which calls for the repetition of 677 the experiment.

How many replicates did you get?

20) In this regard, the two highly expressed BpePIP2s in 695 leaves are the BpePIP2;1 and the BpePIP2;4 (Figure 4), + Figure 4

Where are the expression data from?

21) Samples were collected in triplicate from the roots, young leaves, female inflores-861 cences and xylem (upper stem, about 10th nodes) of healthy two-year-old birch planted 862 in the experimental field of Northeast Forestry University (Harbin, China). Two biological 863 replicates per sample were sequenced, and each biological replicate consisted of a pool of 864 three plant RNAs. 865

Pls, indicate the growth conditions.

22) Concerning the cold stress assay, samples were taken from young leaves from two-866 month-old B. pendula plants grown in the greenhouse at a constant temperature of 25 °C 867 with a photoperiod of 16 h of light and 8 h of dark.

What were the conditions of illumination, the intensity?

23) thus opening up new horizons for very 913 interesting research. 914

915

Correct but it’s good to discuss the results more. Are there are specific differences in aquaporins in the tree compared to e.g. Arabidopsis and the other grasses?

24) Figure S7.

Why are the structures so asymmetric from the point of electric charge, is it so?

25) The submitted Manuscript provides interesting data though mostly lies in the field of bioinformatics with very little amount of own experimental data.

Author Response

Comments and Suggestions for Authors

Definitely, here is an interesting paper with large volume of data and supplementary material. However, the paper still requires essential revisions before being considered further.

1) Quality of figure S8 is low, it’s not readable.

[Authors] Figure S8 readability was improved by increasing resolution to 300 dpi.

2) Figure S12, it’s reasonable to give the full name in the figure legend for TMM values, not the abbreviation initially.

[Authors] TMM full name has been added as appropriate in the FigureS12, and in the list of the “Supplementary Materials” item that appears in the main text.

3) Table S1, FXXK01000578.1a , FXXK01000578.1b , FXXK01001578.1b , FXXK01001578.1a , FXXK01001578.1c , FXXK01002938.1a , FXXK01002938.1b are not found in ncbi. Pls. check and correct, the nucleotides without letters are OK.

[Authors] All the numerical codes used for this study were derived from the NCBI nomenclature with the following rationale.

The Betula genome is mostly sequenced, but the coding sequences (i.e. putative genes) that compose it are not yet annotated. To link contig sequence to annotations, each accession corresponds to a GSS "contig" sequence (whole genome shotgun sequence), which, possibly, may include one or more genes. The letter we subscripted to this code indicates that several aquaporin sequences are found on the same GSS contig.

We added these requested information in results, (lines 135-136) and Material and methods (lines 868-871) as following: “Each BpeAQP gene accession is derived from the identifier of the GenBank GSS contig on which it is located. When necessary, it is subscripted with an additional lowercase letter to indicate that BpeAQP genes accessions differing only by this letter were originally assembled in the same GSS contig.”.

4) Pls, read and consider to cite + more papers from the book:

Gambetta, G.A.; Knipfer, T.; Fricke, W.; McElrone, A.J. Aquaporins and root water uptake. In Plant Aquaporins. Signalling and Communication in Plants; Chaumont, F., Tyerman, S., Eds.; Springer: Cham, Germany, 2017; pp. 133–153.

[Authors] We improved introduction (lines 122-129) by describing how cold stress could be related to global dehydration, considering the reference suggested by reviewer #2 and developing this point including references 38-39-40 to clarify this point.

5) Pls, refer to a few first papers on aquaporins dated back to1980s-1990s with functional characterization (e.g. in frog oocytes), the citation list is skewed for recent papers of the last years.

[Authors] These historical references are generally cited in the papers we mentioned, however, we have added three historical references [17, 18, 22] to the MS (line 75, 85)  in order to address this request.

6) a variety of inorganic and organic solutes such as inter 95 alia boron, silicon, ammonia, glycerol, hydrogen peroxide, urea, polyols, glycine

 Silicon and boron are transported in the forms of acids,not the mentioned elements.

[Authors] We corrected the text accordingly (lines 98, 391, 595, 612, 629) as requested.

7) The Introduction is basically good and comprehensive.

8) are under 117 investigation [32-34] where aquaporins are vital regulators of plant-water relations. 118

 The role of aquaporins is actually overestimated. The main points for plant water relations are the water potentials. Pls, have a look at the mentioned paper at point 4.

[Authors] It is true that aquaporins are part of a vast range of molecules belonging to different biochemical families (proteins, sugars, polyols, amino acids for the main ones), responding in interregulated networks to shape the plant water status. However, the hydraulic status of stressed plants is also part of the whole response of the plant to water deprivation and is impacted after minute cellular responses, some of which being triggered by gene expression and transporters, such as aquaporines, impacted by the physico-chemical status of the plasmalemma membrane. Deciphering causality in these inter-regulated networks is still in its infancy and hydraulic response is only part of the answer where ecophysiological experimentations are mandatory with this respect. However, we addressed this interesting remark from reviewer #2 by outlining this complexity in the text (lines 122-129), but we consider developing it further with bibliographic citations would increase speculations too much in the MS and would be quite beyond its scope.

9) The genomical, structural and biochemical features were determined on 123 the entire set of BpeAQPs, therefore classifying each sequence in the general plant AQP 124 system.

 Not determined but predicted based on the present software, the data were not obtained/confirmed experimentally.

[Authors] It is true that we inferred results from raw data already published but well undermined until this work. We added therefore expertized information to the general corpus of knowledge for aquaporines integrating BpeAQP in that frame. We corrected the sentence as following: “The genomical, structural and biochemical features were inferred, based on present knowledge, for the entire set of BpeAQPs published here from raw and unannotated sequences from NCBI databases, therefore classifying each sequence in the general plant AQP system.” (lines 135-136)

10) Table 1, not readable well. Pls, correct. Especially, the first, last and TMH columns.

[Authors] This Table 1 was completely rebuilt for improved readability.

11) Figure 1.

 The numbers are not provided for all the pairs. Pls, add or discuss.

[Authors] Figure legend already indicates that bootstrap values below 50% are filtered out. This explains why not all pairs do have an attached value on the figure. In phylogenetic tree, bootstrap values below this limit are classically considered as uninformative and are classically not discussed further.

12) Same as 11) for figure 2.

[Authors] See answer 11 this above.

13) Figure 1, there are 34, not 33 aquaporins depicted at the picture. Pls, check all and correct somewhere (text or figure) with XIP2-like (preferably before the figures 1, 2).

[Authors] The analyses presented in the Figure 1 and 2 include the pseudogen BpeXIP2-like because its features are discussed in the text. We have added this specific information in the captions (lines 202, 264).

14) Table 2. the first two columns are not readable. Pls, correct.

[Authors] The Table 2 was rebuilt to improve readability.

15) need to be evaluated further, particularly for the BpePIPs and 359 BpeTIPs, by adopting appropriate computational models [62].

 Need crystal structure and experimental results, not computer models.

[Authors] We acknowledge that an experimental structure obtained on crystallized pure proteins is a must to ground molecular dynamics 3D structures. Progress made only since few years place now 3D folding algorithms in a more favourable position nowadays in terms of accuracy and reliability (see recent CASP challenges). However, Betula pendula studies are still very exploratory and calls for dramatic research efforts to reach the goal of producing purified AQP crystals for RMN studies which are reserved until now to widely studied plant model species (i.e Arabidopsis, spinach, grapevine). Building these efforts requires studies and expertise in order to challenge existing published data. This is the aim of this paper. The goal of publishing a reference crystal AQP from Betula pendula pure protein is well beyond the scope of this paper.

The sentence has been corrected as following: “//need to be evaluated further, particularly for the BpePIPs and BpeTIPs, by adopting appropriate computational models [68], involving finally, functional and structural validations.“. (lines 387-388)

16) With the noticeable exception of holoparasitic plants, CO2 435 is considered as an universal substrate for plants, and the facilitation of its membrane 436 diffusion at the mesophyll seems to be exclusive of the plant PIPs

 Not clear about universal substrate, too philosophical. Pls, rephrase in a clear form.

[Authors] This sentence has been rewritten accordingly: “With the noticeable exception of holoparasitic plants, CO2 is considered as a key substrate for plants (i.e. photosynthesis), (lines 474-475)”

17) All this together opens the possibility that some 455 PIP2s could diffuse CO2 in concert with certain PIP1s in B. pendula.

Not correct phrase, PIP2 can not diffuse, CO2 can diffuse.

[Authors] This sentence has been rewritten accordingly: ”All this together opens the possibility that CO2 can diffuse through some PIP2s in concert with certain PIP1s in B. pendula.” (lines 494-496)

18) The specific membrane localization can vary between the different AQP subfamilies, 636 which, ultimately, influences sub-cellular flow and compartmentalization of solutes. Al-637 most all of the BpeAQPs are predicted to be localized to the plasma membrane, except for

 Not correct phrase, see your own text earlier for localization and its description.

[Authors] Our argument has been rewritten accordingly: “The plant aquaporins are mainly located in the plasma membrane. However, specific membrane localizations can vary between the different AQP subfamilies, ...” (line 679)

19) However, although several trends in trimmed mean of M-675 values (TMM) can be distinguished, they are not statistically significant (p>0.05) (Figure 676 S12) mainly due to the paucity of biological repetitions, which calls for the repetition of 677 the experiment.

 How many replicates did you get?

[Authors] The number of replicates was detailed in this sentence: “However, although several trends in trimmed mean of M-values (TMM) can be distinguished, they are not statistically significant (p>0.05) (Figure S12) mainly due to the paucity of biological repetitions (i.e. 2 or 3 biological replicates depending on the kinetic point), which calls for the repetition of the experiment.” (lines 722-725)

20) In this regard, the two highly expressed BpePIP2s in 695 leaves are the BpePIP2;1 and the BpePIP2;4 (Figure 4), + Figure 4

 Where are the expression data from? BpePIP2;1 and -;4 correspond to

[Authors] The expression levels of BpePIP2;1 and -;4 can be seen at the 6th and 7th AQP lines of the 1st part of the graph which is entitled “leaf” (top left).

21) Samples were collected in triplicate from the roots, young leaves, female inflores-861 cences and xylem (upper stem, about 10th nodes) of healthy two-year-old birch planted 862 in the experimental field of Northeast Forestry University (Harbin, China). Two biological 863 replicates per sample were sequenced, and each biological replicate consisted of a pool of 864 three plant RNAs. 865

 Pls, indicate the growth conditions.

[Authors] Growth conditions were completed in corresponding material and methods section (lines 939-942).

22) Concerning the cold stress assay, samples were taken from young leaves from two-866 month-old B. pendula plants grown in the greenhouse at a constant temperature of 25 °C 867 with a photoperiod of 16 h of light and 8 h of dark.

 What were the conditions of illumination, the intensity?

  [Authors] Illumination conditions were added in the corresponding materials and methods section (lines 945-946). However, the intensity light flux was not measured during this experiment.

23) thus opening up new horizons for very 913 interesting research. 914 915

Correct but it’s good to discuss the results more. Are there are specific differences in aquaporins in the tree compared to e.g. Arabidopsis and the other grasses?

[Authors] There are many transcriptional studies that decipher the molecular actor that drive the cold stress response, and some -recent- reviews provide a panorama of how some aquaporins are modulated during cold stress (are mainly concerned: PIP, TIP and NIP) [127]. However, to our knowledge, studies that specifically focus on the modulation of the entire MIP superfamily under cold stress conditions are scarce (eg Musa acuminate [128] and Olea europaea [5]).

We acknowledge that several partial datasets on AQPs do exist from different annual species, such as Arabidopsis [130] or rice, [129] and see references mentioned in the above-mentioned reviews [127]. We have chosen to cite prominently only the studies that dealt with the MIP superfamily in the most complete way possible. However, discussion has been elaborated further (lines 732-739) to address reviewer’s request.

Lastly, it is always possible to compare the expression profile between orthologs; however, this comparison remains very speculative, and even more so when the orthologs come from species that are phylogenetically distant (e.g. Arabidopsis and Birch). We try to be very cautious with this regard.

24) Figure S7.

Why are the structures so asymmetric from the point of electric charge, is it so?

[Authors] It is true that many AQPs are considered as an "electrical dipoles". The reasons for this electrical polarity is not fully understood yet, and still under investigations in our group. It could be related to the electrical polarity of the biomembranes, therefore stabilizing the embedded protein. In addition, this contrast of charges could be key in the regulation of the intrinsic channel activities of the AQPs, knowing that these charges are quantitative and that the pH, phosphorylation, and Ca2+ could modulate them. We modified the corresponding part of the conclusions (lines 396-403) to emphasize this remark from reviewer #2.

25) The submitted Manuscript provides interesting data though mostly lies in the field of bioinformatics with very little amount of own experimental data.

[Authors] We thank you for your encouragements. In silico biology results are produced at an increasing pace since one decade and we are involved in this new multidisciplinary research area. In this paper, we associated in silico biology results to experimental biology results obtained on gene expression in Betula tissues aiming at understanding the activity of some specific AQPs towards their functional validation. We must emphasize that in silico biology results are based on experimental hypothesis and refine to a high level the protein functional and structural annotations of what could remain raw data in the public databases. We are still far from being able to automatize this process for all protein sequences since it still requires a vast amount of human expertise for validation. This human expertise is part of what we propose in the present publication.

Round 2

Reviewer 2 Report

1) Quality of figure S8 is low, it’s not readable.

[Authors] Figure S8 readability was improved by increasing resolution to 300 dpi.

The Reviewer is still not able to see the amino acids mentioned at the figure S8 at the motifs 1-8, 9-10 are OK. Pls, have a respect to the persons who would like to read your MS and make the information visible in any form. Pls, think how to express your interesting and useful information in a reliable and robust way.

2) Figure S12, it’s reasonable to give the full name in the figure legend for TMM values, not the abbreviation initially.

[Authors] TMM full name has been added as appropriate in the FigureS12, and in the list of the “Supplementary Materials” item that appears in the main text.

Still the figure is not impressive. What is T (first letter of the second row) standing for? h and H are for hours according to my humble guess, why not to use one h without a capital H? Any outstanding and useful reasons exist?

3) Figure S2. regularized-log transformed

What is the base for the log? 2, e, 10?

Author Response

1) Quality of figure S8 is low, it’s not readable.

[Authors] Figure S8 readability was improved by increasing resolution to 300 dpi.

The Reviewer is still not able to see the amino acids mentioned at the figure S8 at the motifs 1-8, 9-10 are OK. Pls, have a respect to the persons who would like to read your MS and make the information visible in any form. Pls, think how to express your interesting and useful information in a reliable and robust way.

[Authors] The residues that are statistically under-represented are de facto overwritten in the figures by the over-represented residues, making them unreadable. In order to make each pattern interpretable, we provide with the FigureS8 the quantified and statistical presence of all the residues that compose them.

2) Figure S12, it’s reasonable to give the full name in the figure legend for TMM values, not the abbreviation initially.

[Authors] TMM full name has been added as appropriate in the FigureS12, and in the list of the “Supplementary Materials” item that appears in the main text.

Still the figure is not impressive. What is T (first letter of the second row) standing for? h and H are for hours according to my humble guess, why not to use one h without a capital H? Any outstanding and useful reasons exist?

[Authors] Each point you raise has been corrected and explained in the legend. The "T" has been replaced by C, which corresponds to the control sample (i.e. plants left at 25°C), and the "H/h" has been harmonised by hpt, for hours post treatment.

3) Figure S2. regularized-log transformed. What is the base for the log? 2, e, 10?

[Authors] This is clearly a remnant mistyping left from computer exponential representation. We corrected to the human readable format as “pvalue=1.94 10-22 ” . Thank you for pointing out.